# The impact of exercise on gene regulation in association with complex trait genetics

Nikolai G. Vetr [1] ✉, Nicole R. Gay [1], MoTrPAC Study Group* & Stephen B. Montgomery [1] ✉

Endurance exercise training is known to reduce risk for a range of complex diseases. However, the molecular basis of this effect has been challenging to study and largely restricted to analyses of either few or easily biopsied tissues. Extensive transcriptome data collected across 15 tissues during exercise training in rats as part of the Molecular Transducers of Physical Activity Consortium has provided a unique opportunity to clarify how exercise can affect tissue-specific gene expression and further suggest how exercise adaptation may impact complex disease-associated genes. To build this map, we integrate this multi-tissue atlas of gene expression changes with gene-disease targets, genetic regulation of expression, and trait relationship data in humans. Consensus from multiple approaches prioritizes specific tissues and genes where endurance exercise impacts disease-relevant gene expression. Specifically, we identify a total of 5523 trait-tissue-gene triplets to serve as a valuable starting point for future investigations [Exercise; Transcription; Human Phenotypic Variation].

Endurance exercise is associated with multiple positive health outcomes[1,2]. However, the molecular basis of these positive effects has been challenging to study, with past work restricted to molecular assays in either few or easily accessible tissues[3]. Even when prior differential analyses have identified exercise-responsive genes, there is often limited evidence for their shared molecular impact on disease. To address this challenge, we have combined the extensive, multi-tissue transcriptome data from the Molecular Transducers of Physical Activity Consortium (MoTrPAC) preclinical endurance exercise training (EET) study in rats[4] with data from the Genotype-Tissue Expression (GTEx) project, where genetic differences in expression levels have been previously connected to 114 traits and diseases from publicly available Genome Wide Association Studies (GWAS) distributed across several phenotypic and plausibly exercise-responsive categories[5] (note: acronyms and abbreviations used in this paper are summarized in Supplementary Table 1). The MoTrPAC EET study provided differential expression results after treadmill exercise training for both female and male F344 rats, with multiple tissues harvested at 1, 2, 4, and 8 weeks of training. All samples were harvested 48 h after the last

exercise bout, and the 8-week time point was taken to correspond to the adapted state, as it allowed for the greatest degree of long-term adaptation to exercise to have occurred, as well as the least degree of unadapted acute response (eg inflammation). In rats, as in humans, exercise capacity is a genetic trait with well-studied relationships across a range of human-relevant complex traits and diseases[6-8]. Combined, these data provided a cross-tissue, whole organism molecular view of adaptation to exercise that is unattainable in human participants.

To assess the relationship of exercise adaption and complex disease risk in distinct tissues, we applied a combination of heritability and transcriptome-wide association study (TWAS) analyses (Fig. 1a). These analyses are state-of-the-art in human genetics but have yet to be broadly applied cross-species in the context of exercise adaption. They allow us to investigate exercise adaptation genes and gene sets for their relationship to specific complex diseases. We applied LDSC[9], which can accommodate linkage disequilibrium to estimate SNP-heritability captured by sets of exercise adaption genes ($h^2_{SNP}$)[10], alongside MESC[11], which incorporates both GWAS and Expression

[1]Stanford University, Stanford, CA, USA. *A list of authors and their affiliations appears at the end of the paper. ✉e-mail: nikgvetr@stanford.edu; smontgom@stanford.edu

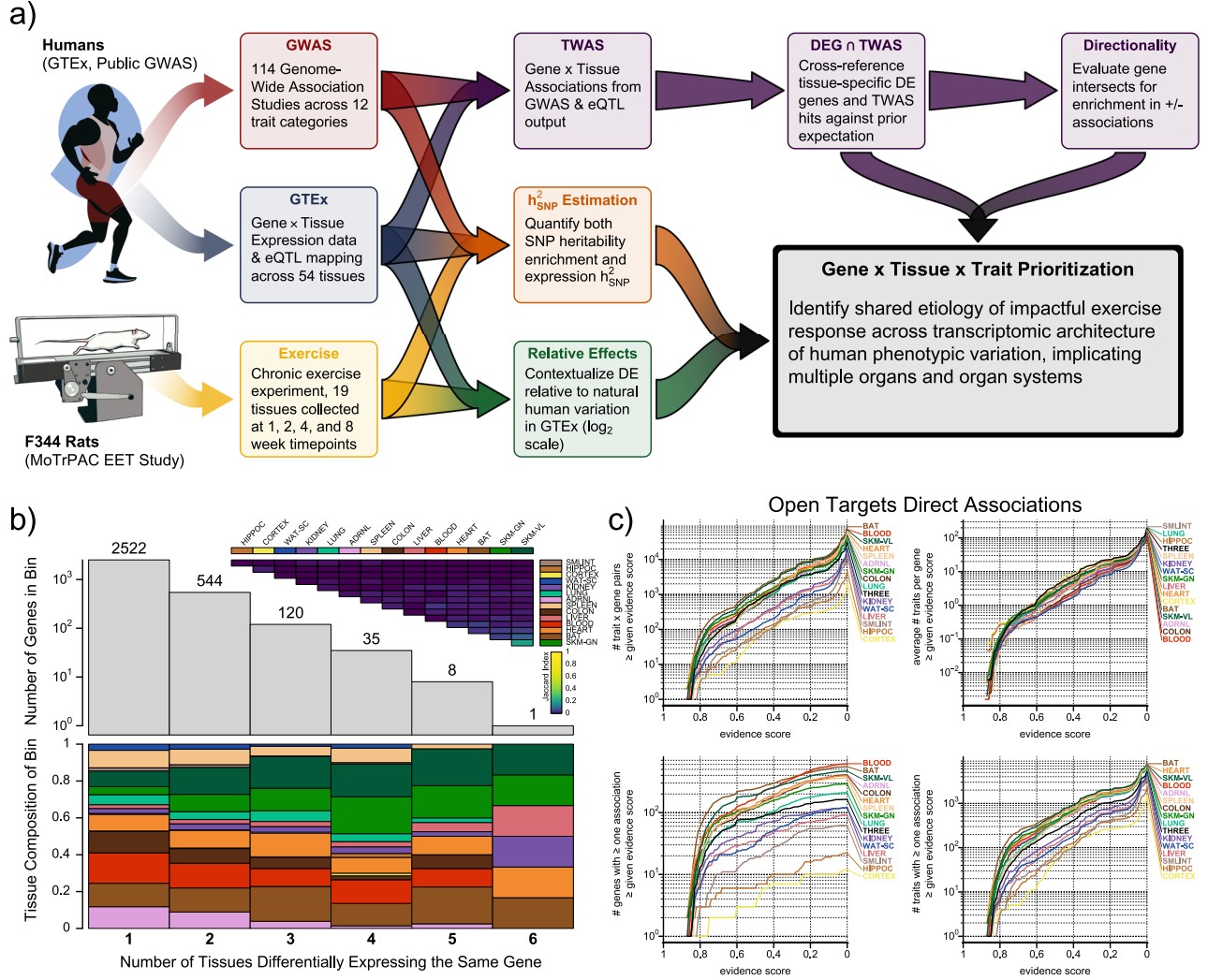

**Fig. 1 | Tissue-specific differential gene expression from exercise impacts unique sets of disease processes.** In (**a**), we provide a general overview of the work described here, from human genetic and transcriptomic data and rats subjected to an Endurance Exercise Training (EET) experimental perturbation to triplets of causally entangled genes, tissues, and traits. In (**b**), we subset Differentially Expressed (DE) Genes to just those determined to be DE at 8W in a sex-consistent

manner, and visualize their distribution and tissue-specific composition across uniquely DE genes (genes DE in only one tissue), pairs (in two tissues), triplets, etc. To check for overlap in these gene sets, we also plot the upper triangle of a Jaccard Similarity matrix. In (**c**), we present alternative ways to characterize Open Targets associations across these gene sets.

Quantitative Trait Loci (eQTL) summary statistics to estimate the proportion of $h^2_{\mathrm{SNP}}$ mediated by gene expression within and across tissues to assess the relationship between genetic variability and adaptive exercise training response. Finally, we leveraged published S-PrediXcan[12,13] output, which estimates gene by tissue-level associations and directions of effect for specific diseases to identify genes where changes in gene expression due to exercise adaptation have the potential to alter disease risk.

Using these data and approaches, we identify gene and tissue combinations where expression levels could mediate disease risk and where exercise training had the potential to induce expression differences capable of overwhelming the impact of human standing variation measured in the GTEx study, both overall and with respect to its genetic component. We further assess if specific diseases and traits are enriched for genes differentially expressed in exercise training, both in their overall occurrence and in their directionality of effect. Combining these approaches, we identify specific genes that lie at this intersection of biological relevance as candidates where exercise effects could override expression-mediated disease risk.

## Results

### Exercise training has unique disease gene signatures across tissues

Exercise training induces differential expression of rat genes across multiple body tissues, and many of these genes can be mapped to human orthologs: 94.5% of all unique, differentially expressed (DE) genes (87–98% across tissues), and 79% of all expressed genes (85–93% across tissues). However, most of these changes exhibit marked tissue specificity. As observed in the main MoTrPAC PASS1B paper[4], we found that after long-term exercise training, there was limited overall concordance of adaptive differential expression across tissues in the subset of rat genes with identifiable human orthologs (hereafter 'genes', unless otherwise noted). Only two pairs of tissues in females—the skeletal muscles *vastus lateralis* and *gastrocnemius*, as well as white adipose and the colon—produced Spearman's '$\rho$'s at a level greater than 0.3 (Supplementary Fig. 1b). Further, there was little overlap in differentially expressed gene sets (DEGs) corresponding to each tissue (Fig. 1b). Approximately 78% of DE genes were unique and differentially expressed in only one tissue, and 95% of genes matched at most a pair of tissues. Only one pair of tissues showed a Jaccard index > 0.1

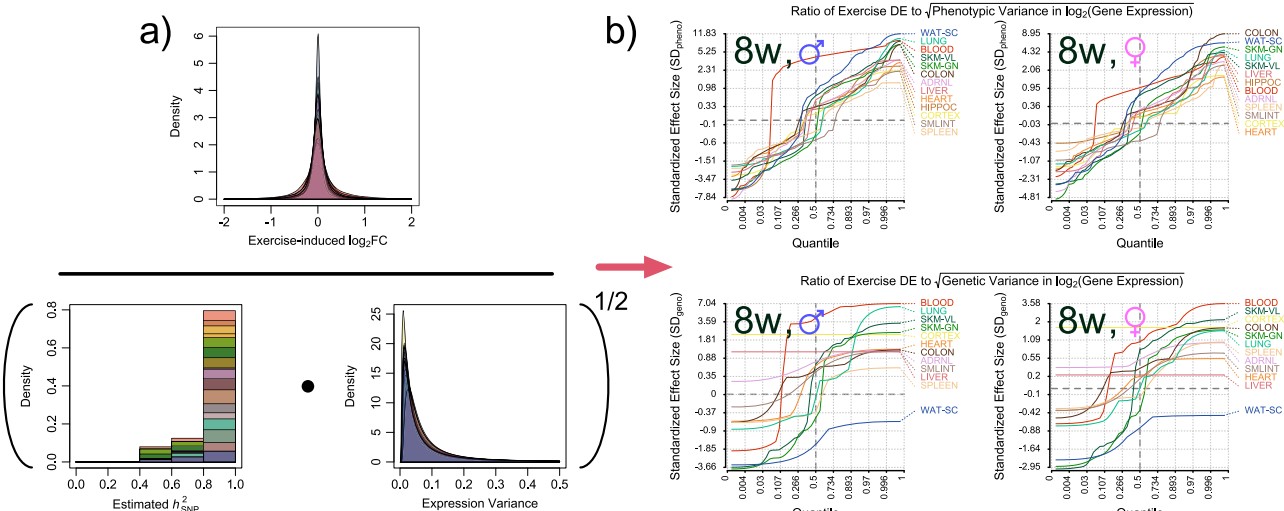

**Fig. 2 | Tissue-specific differential gene expression from exercise can exceed natural variation.** In this figure we visualize the procedure used to obtain Standardized Effect Sizes. In the numerator of (**a**) lies a truncated kernel density estimate of the distribution of $\log_2$FCs induced by exercise at the 8w timepoint. A stacked histogram of estimates for $h^2_{\text{SNP}}$ for expression scores from GTEx ($p < 0.10$ after IHW correction) is on the left in the denominator. On the right are the inverse-gamma distributions serving to regularize $\log_2$-normalized expression scores. Together, this results in a value equal to the estimated genetic variance of expression. Taking its square root yields a standard deviation, which we use to divide exercise-induced $\log_2$FC. In (**b**), we plot the empirical quantile function for

distributions of ratios of each tissue's exercise-induced $\log_2$(gene expression) / SD($\log_2$(gene expression)). As most of the interesting behavior is contained in the tails of each distribution, we applied two separate transforms to the axes of each plot. The horizontal axis, corresponding to a given quantile in (0,1), was logit-transformed. The vertical axis, corresponding to the ratio of DE / SD($\log_2$expression), received an inverse hyperbolic sine transformation. The upper panels are in units of standardized phenotypic effect, and the lower in units of standardized genetic effect for those genes and tissues with significant non-zero $h^2_{\text{SNP}}$ (IHW $\alpha = 0.10$, one-sided). Source data for this figure are provided as a Source Data file.

(the *gastrocnemius* and the *vastus lateralis*, Jaccard Similarity ≈ 0.21). This indicates that unique genes and pathways adapt to exercise training in different tissues and likely impact different subsets of disease-relevant genes. To this point, we observed 370 high-scoring (Open Targets evidence score > 0.8, an arbitrary threshold chosen to select gene × trait relationships with high levels of supporting evidence) disease genes from 251 traits across our 15 surveyed tissues (Fig. 1c) that were consistently responsive to exercise training in both males and females, with an average of 18.2 genes per tissue. When we excluded easily biopsied tissues such as blood, skeletal muscles, and adipose, we found 178 well-established disease genes associated with an adaptive exercise training response. This corresponded to 143 traits and included 101 traits without any gene-trait associations in an easily biopsied tissue. Notably, these included well-studied genes such as *LDLR* (DE in {CORTEX, HIPPOC, SKM-GN, SKM-VL} and *APOB* (DE in {COLON, KIDNEY, LUNG}), both confidently associated with hypercholesterolemia; *SLC6A8* (DE in {HEART, LIVER, LUNG}), associated with creatine transporter deficiency; *FOXP3* (DE in {HEART, SPLEEN}, associated with immune dysregulation; and *BRCA2* (DE in ADRNL), associated with breast neoplasia.

### Exercise effects on regulation of gene expression

We sought to identify where changes in gene expression due to exercise training could potentially overcome either genetic or standing variability measured in GTEx. Here, our hypothesis was that exercise behavior may be more impactful than baseline variance or genetics at these loci for the component of disease risk mediated by gene expression. For each gene and tissue where we could detect non-zero genetic $h^2_{\text{SNP}}$ (IHW $\alpha = 0.10$), we calculated genetic variance as the product of the estimated heritability and observed total phenotypic variance (Fig. 2). At 8W, we observed an average of 1 (range: 0–10) genes per tissue in at least one sex with effect sizes in trained rat that were > 2SD the genetic component of expression variability of the matched sex in humans (SD$_{\text{geno}}$), and 52 (range: 1–586) genes per tissue

with DE > 2SD overall expression variability (SD$_{\text{pheno}}$), with the latter set featuring ≈ 50 genes per tissue whose $h^2_{\text{SNP}}$ could not be significantly distinguished from 0 after multiplicity adjustment. Intersecting these genes with Open Targets, we observed 30 unique > 2 SD$_{\text{pheno}}$ DE genes with > 0.8 evidence scores, though only 14 of these were expressed in less accessible tissues. *APOB* was included in the latter group, mentioned above (DE in male *lung* at ≈ +9.7 SD$_{\text{pheno}}$, and in the female *colon* at ≈ +2.4 SD$_{\text{pheno}}$).

### Heritability of complex disease enriched in or near training-responsive genes

We investigated whether exercise specifically modulates any traits or diseases by building on a previous approach[14] to identify these effects. First, we computed the trait or disease heritability for gene sets that were differentially expressed due to exercise training in each tissue at 8W in both sexes and in the same direction. We observed the strongest magnitude of enrichments in *blood* phenotypes in the blood tissue, especially traits corresponding to densities of immune cells (Fig. 3).

Across the 43 traits with at least one significant enrichment at Bonferroni-adjusted $\alpha = 0.05$, the largest significant enrichment factor corresponded most often to the spleen (22/43 ≈ 51%), especially in *Blood* (9/14) and *Immune* (6/7) phenotypes, with an average enrichment factor of ≈ 2.85 across significant enrichments. The proportion of heritability captured by these gene sets is on the order of ≈ 10% (Supplementary Fig. 2a) and corresponds to broadly independent signals across tissues (Supplementary Fig. 2b–c). This approach provides a general prioritization for assessing which traits or diseases could be most impacted by exercise training. However, we performed simulation experiments using randomly sampled gene sets of equivalent size to our original tissue-specific gene sets. These produced highly similar distributions of *p*-values to those observed for the empirical data. As such, these results (Fig. 3) should be interpreted less in the framework of null-hypothesis significance testing and more descriptively, as a relative ordering of estimated magnitudes of effect.

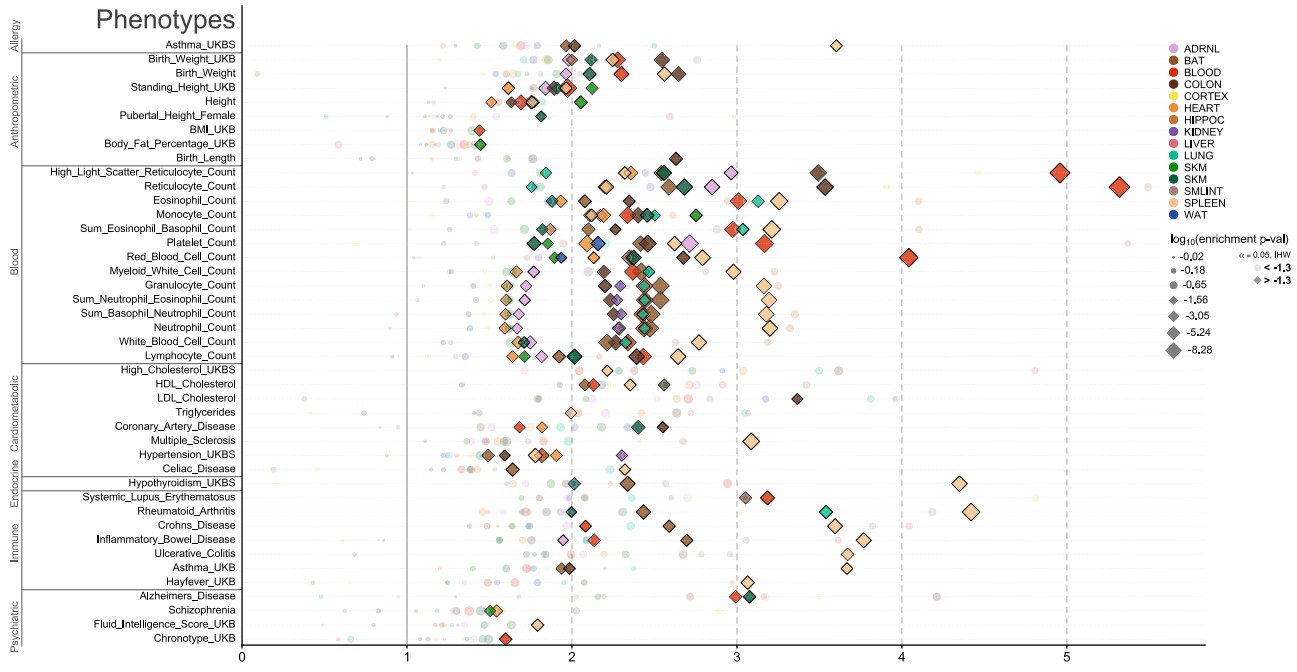

**Fig. 3 | Genetic variation near exercise training genes is enriched in heritability across human phenotypes.** Here, we visualize conditional heritability enrichments (LDSC) of multiple traits within differentially expressed, sex-independent gene sets corresponding to different tissues. Colors distinguish tissues, with opaque diamonds corresponding to IHW-significant hits ($\alpha = 0.05$), and size proportional to the magnitude of $\log_{10}(p\text{-value})$ (two-sided, adjusted for multiple comparisons with IHW). The horizontal axis corresponds to the heritability enrichment factor, and the vertical to GWAS traits, grouped into high-level categories. Traits lacking an IHW-significant ($\alpha = 0.05$) hit in at least one tissue are excluded from this visualization, and the horizontal axis has been truncated to exclude non-significant enrichments above that of the maximum significant enrichment, as well as estimated enrichments < 0, which are strictly impossible. Source data for this figure are provided as a Source Data file.

## PrediXcan-significant genes overlap adaptive training-response genes

We examined the intersection of genes that are differentially expressed at "8w_F1_M1" and "8w_F-1_M-1" (i.e., sex- and direction-consistent after 8 weeks of training) and IHW-significant PrediXcan hits (Fig. 4a). We were able to identify substantial enrichment in many of the traits, trait categories, tissues, and tissue-by-trait pairs through the use of a hierarchical Bayesian model able to partially pool estimates of difference effects towards the means of their respective populations. Here, we see confident (>95% posterior probability) enrichments across all levels of the model hierarchy (Fig. 4e–g). Specifically, we observed confident positive differences in the colon, kidneys, small intestines, spleen, hippocampus, lungs, and heart, in order of decreasing posterior mean, as well as in the Endocrine and Cardiometabolic categories. We also noted several specific trait enrichments across cardiometabolic markers, mainly cholesterol and saturated fatty acids. At the trait × tissue level, posterior output were broadly uncertain in most pairs' directionality of enrichment, with a smaller number showing stronger confidence in positive enrichment (Fig. 4b).

Conversely, none of the frequentist analyses of this overlap produced significant results at FWER $\alpha = 0.05$ ($-\log_{10}(0.05) \approx 1.30$, one-sided), with the most significant result corresponding to the multi-tissue GSEA for high cholesterol at an adjusted $p$-value of ≈ 0.064 (Supplementary Fig. 3, ES ≈ 0.63, log2err ≈ 0.48). However, results were broadly concordant across the two approaches, and more confident posterior distributions corresponded to lower frequentist $p$-values, with intermediate positive Spearman's $\rho$s for pairwise and trait-wise comparisons (Supplementary Fig. 3a–c). Frequentist meta-analysis of tissue and trait-category enrichments were in less confident agreement, with the latter showing mild disagreement, though at $p \approx 0.53$ (output from `stats::cor.test` in $R$).

## Exercise induces both more and less disease-like differential gene expression

To identify the direction of training effect in these intersecting gene sets, we queried the posterior output from a second multilevel model, visualizing posterior means for each tissue and tissue-by-trait combinations as a dot plot (Fig. 5). Given the reduced capacity for signal in these data (focal totals no longer being the set of DEGs, but the set of DEGs ∩ PrediXcan hits), we report on confident effects when a posterior mass is >90% to one side of 0. As such, the strongest confident mean enrichments for positive effects were observed in body fat percentage, asthma, and body mass index (BMI), and the strongest mean depletions in standing height and high cholesterol, though of the latter only standing height was "confident". Otherwise, body fat percentage was the only trait with posterior difference >95% in either direction. As traits varied in the degree to which they could be considered harmful or beneficial, we could not evaluate gross tissue effects across traits, but at the tails of each trait's hyperdistribution, blood, spleen, and the two skeletal muscles had the strongest degree of deviation from null expectation. Additionally, ≈83% of the posterior mass of our $G_{SNP}$ weight parameter $\theta$ fell above 0.5, with ≈28% falling above 0.9.

When examining the direction of trajectories for 8-week gene sets linked to the two non-anthropometric traits, we noticed a regression towards a mean proportion of 0.5 across tissues. This is likely due to underlying genes only being differentially expressed at later time points (Fig. 6). Examining which genes and tissues correspond to both high deviation from the mean and relatively large amounts of DE, we observed blood genes associated with lower cholesterol in males (*NDUFA13*, *FADS2*, *PNKD*, *AAMP*, and *OGDH*), as well as the male *vastus lateralis* gene *TMBIM1*, the female-specific training gene *APOB* in colon, and the female training gene *ABCG8* in liver. With respect to increased risk of asthma, blood genes again had the largest

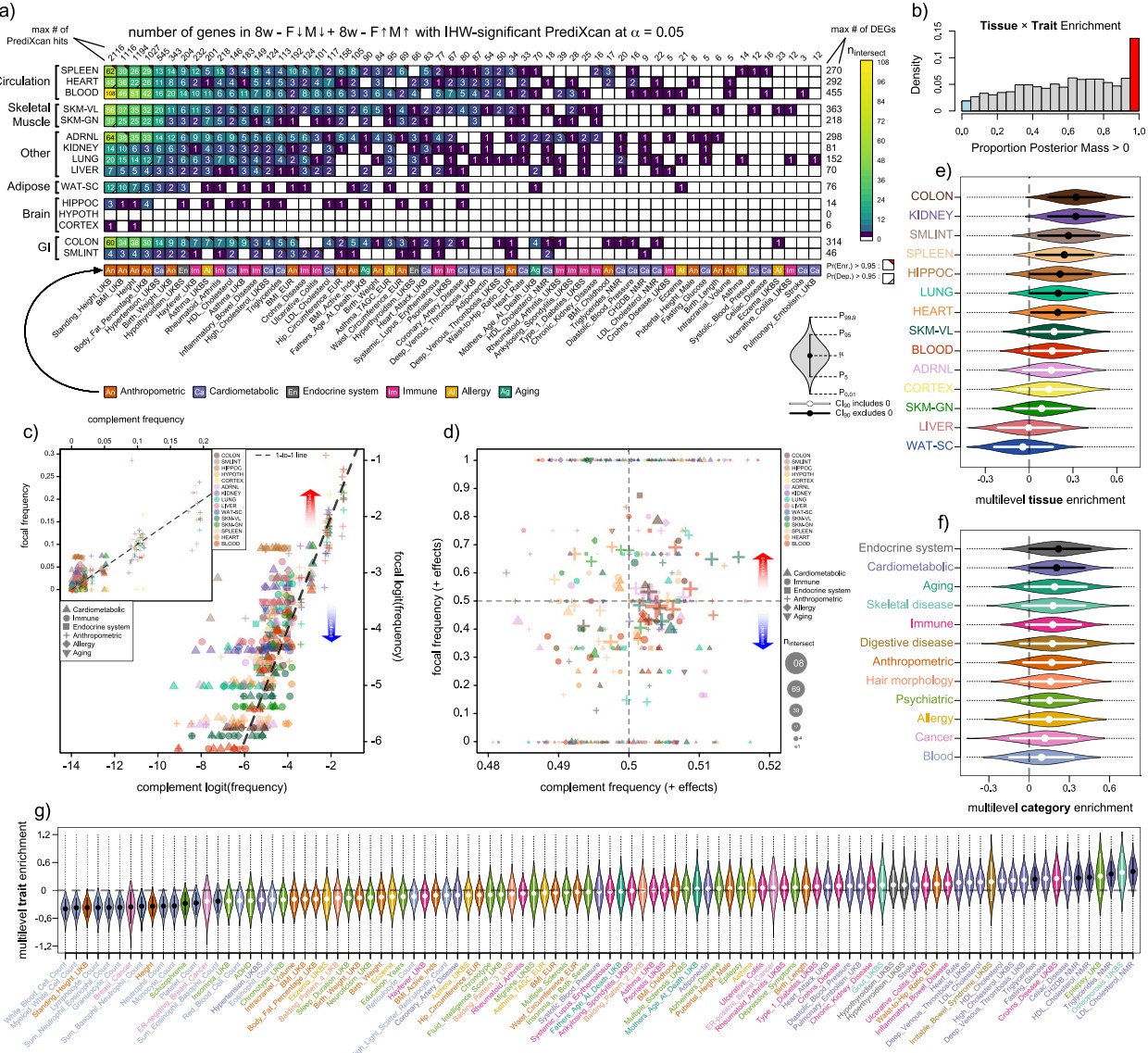

**Fig. 4 | Exercise training genes are enriched for genes where expression is associated with trait variation across multiple trait categories.** Here, we visualize fitted output from our PrediXcan-DEG intersect enrichment model (n = 10,000 nominal iterations across four independent chains). In (**a**), we show the sizes of gene sets in the intersect of PrediXcan hits (IHW $\alpha = 0.05$) across different traits (horizontal axis) and sex-homogeneous, differentially expressed genes (DEGs) at 8W in different tissues (vertical axis). Cell colors correspond to the size of the intersecting gene set. Numbers in each cell give the size of each intersect, with cell corners labeling cells whose marginal posterior difference parameter has >95% of its mass to one side of 0. Marginal counts give the maximum number of PrediXcan hits in each trait (vertical margin) or DEGs in each tissue (horizontal margin), after constraining the total pool to mutually expressed genes. In (**b**), we plot the histogram of posterior masses > 0 for trait × tissue difference effects, with colors drawn from cell labels in (**a**). In (**c**), the vertical axis corresponds to logit-transformed frequencies of PrediXcan hits in the DEGs from (**a**), and the horizontal

axis represents the corresponding frequency in all genes outside this set. Only traits from six trait categories are depicted, with colors corresponding to tissues and shapes to categories. In (**d**), the vertical axis maps to the proportion of positive effects in the PrediXcan-DEG intersect across traits and tissues, and the horizontal axis to the same proportion outside that intersect. Point diameter is proportional to the square root of intersect gene set size, with colors and shapes retaining their meaning from (c). In (**e–g**), marginal posterior distributions from our intersect enrichment model are shown as violin plots, with internal lines representing middle 90% credible intervals and internal points representing posterior means. Internal line and point colors are white when the credible interval overlaps with 0, and black otherwise. Violins are arranged in order of increasing posterior mean, with the horizontal axis on the logit scale. In (**e**), we plot these at the tissue level, in (**f**) at the trait category level, and in (**g**) at the trait level. Source data for this figure are provided as a Source Data file.

relative effect sizes in males (*BAG6*, *CCNG*, *CRAT*, *PTPA*, and *FAM89B*), with female training genes exhibiting the largest effects in *ATP6V1G2* in the *vastus lateralis*, *ENDOU* in white adipose, and *CCNF* in blood.

## Discussion

In our study, we have identified multiple tissues and tissue-by-gene pairs where exercise may modify disease risk through gene expression. Despite human-rat differences, our unbiased approach identified

multiple results that echo established exercise-disease relationships. However, some findings were unexpected.

Gene sets that responded to exercise were enriched for PrediXcan genes linked with cardiometabolic traits (Fig. 4e, f). The intersection of these genes seems to lean away from disease-like effects (Fig. 5), but we also found disease-like effects for genes associated with asthma and body fat percentage (Fig. 5). These associations, however, did not exhibit intersect sizes larger than expected by chance, and the latter

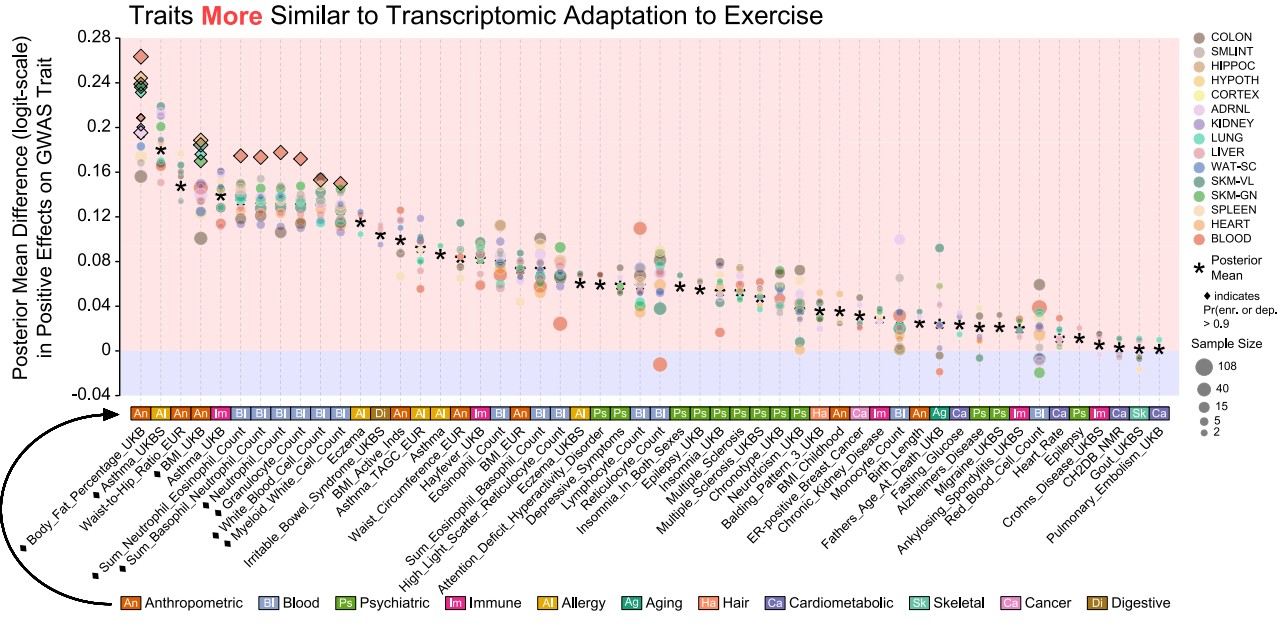

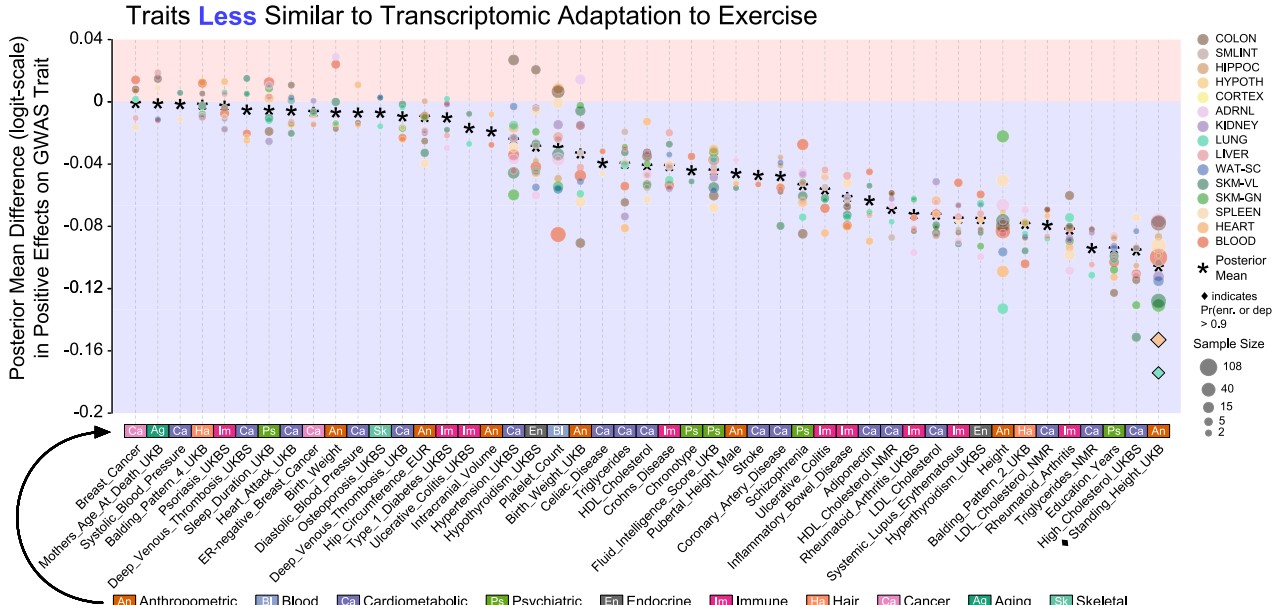

**Fig. 5 | Exercise training genes can be enriched for more or less disease-like effects.** We visualize the posterior means of multilevel trait and trait × tissue terms from a Bayesian model corresponding to the proportion of genes imparting a positive effect on traits aligned along the horizontal axis. Diamonds mark traits or trait × tissue pairs whose difference effect's posterior mass falls entirely to one side of 0, either prepending the trait name or else marking the trait × tissue symbol. Points sizes are in proportion to the square root of sample size, and traits are arranged on the horizontal axis according to the monotonic decrease in their posterior means. Source data for this figure are provided as a Source Data file.

showed only weak evidence of depletion (Fig. 4f). Additionally, when aggregating across traits, several of the most "classically" exercise-responsive tissues – the skeletal muscles and white adipose–appeared to be among the most depleted for PrediXcan hits (Fig. 4d), though no marginal posterior distributions for difference parameters there reached our 95% posterior mass threshold. Overall, estimates for enrichments and depletions in both intersect and directionality effects were small, even for confidently non-zero effects, predominantly varying within $0 \pm 0.3$ on the log-odds scale (Figs. 4e–g, 5). This corresponds to a maximum difference of ≈7.5% on the probability scale (`inv-logit(0.15) - inv-logit(-0.15)`), and is consistent with the relatively small deviations observed from the 1-to-1 lines in Fig. 4c–d. Interpretation of these exercise biological findings should not lose

sight of this context: small, subtle, but nevertheless discernible association.

In the case of body fat percentage (BF%), it may be that absent dietary control – for example, when rats are fed *ad libidum* – genes are regulated in a manner that elicits increased fat storage as an adaptation to higher energy expenditure[15]. Thus, even though exercise may often be done by humans with the goal of reducing BF% through increased caloric expenditure, an interaction with diet modulates this response. However, white fat itself does not appear to be enriched for positive effects on BF%, with the most pronounced enrichments evident in blood and heart. Similarly, the strongest depletion for positive effects, both within anthropometric traits and overall, occurred in the standing height phenotype. Evidence for exercise effects on height is weak and

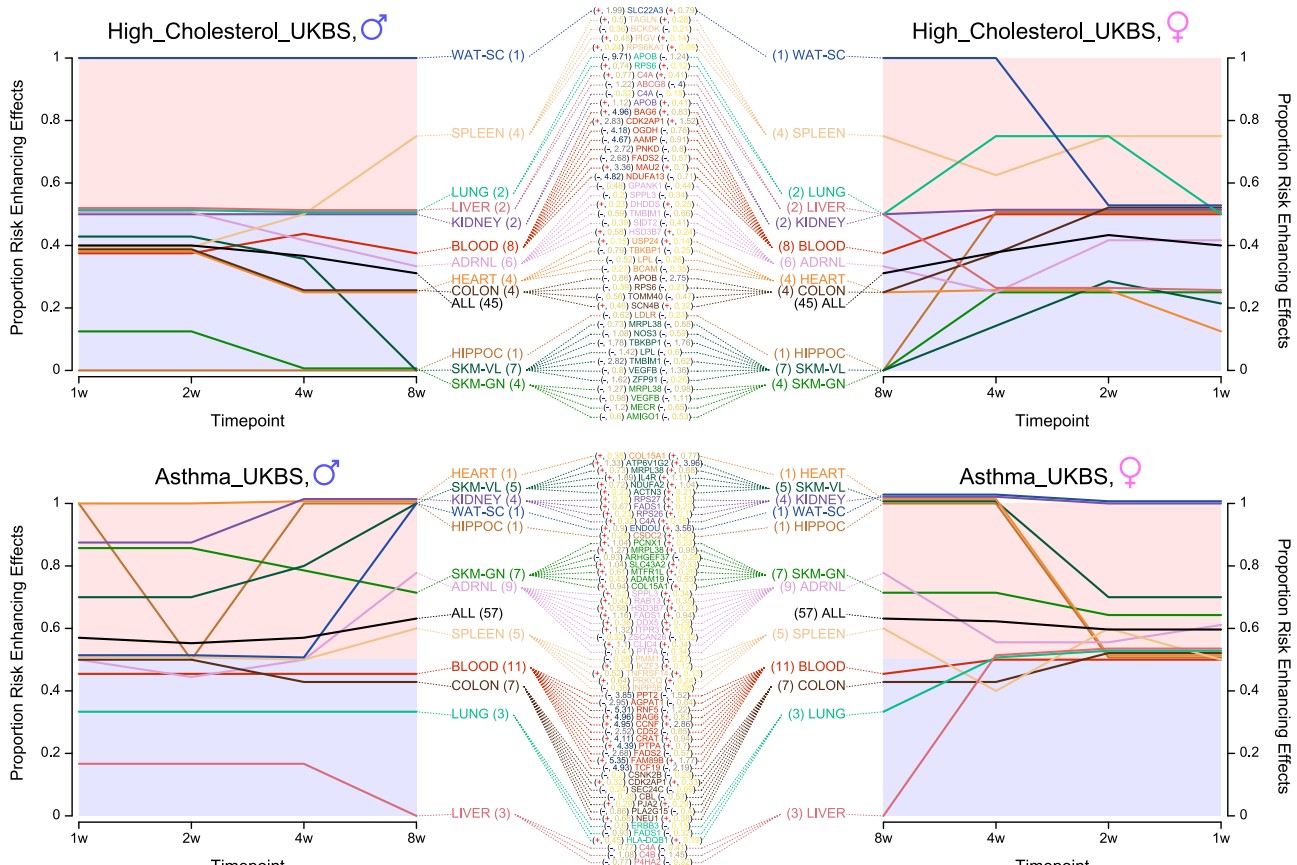

**Fig. 6 | Examining which trait-associated, exercise-responsive genes are differentially expressed at levels outside natural variation yields interesting candidates for further study.** We visualize the observed proportion of positive effects for the two non-anthropometric traits that had the highest posterior mean enrichment in that proportion according to our proportion of positive effects enrichment model. Above, two panels correspond to self-reported high cholesterol, and below, self-reported asthma. Lines terminate at 8W on the right of each panel, splitting into tissues and genes. Additionally, we trace the proportion for the 8w gene set backwards in time, examining the effect of those genes at 1w, 2w, and 4w. Tissue names are followed by the total number of genes in the intersecting gene set in parentheses, and gene names are followed by their sign (a red + if the effect of DE on the trait is positive and a blue − if negative), and the standardized effect size of DE from Fig. 2b. Additionally, we plot a line corresponding to the set of all gene-tissue pairs in black, labeled ALL. Source data for this figure are provided as a Source Data file.

ambiguous. However, particularly intense exercise may have an attenuating effect on growth[16–18], especially under nutritional stress, which may partially underlie the associations observed here.

Asthma also emerged as a trait with shared transcriptional effects as exercise. This may be due to similar etiology between the general asthmatic condition measured by self-report and exercise-induced bronchoconstriction (EIB), where lung epithelial stress from exercise and increased drying and cooling of the airways due to increased ventilation triggers an inflammatory response alongside shortness of breath[19]. Though no individual tissues were found to be confidently enriched in positive effects for this phenotype (Fig. 5), DE genes in the spleen—a key immune and inflammatory response regulator[20] – emerged as having the greatest enrichment in $h^2_{SNP}$, comprising nearly four times baseline expectation and accounting for ≈10% of trait heritability overall (Fig. 3, Supplementary Fig. 2a). Moreover, DE genes in the spleen had the highest $h^2_{SNP}$ enrichment across a number of additional immune cell and disease phenotypes, and both eosinophil and basophil counts were found to have moderate genetic correlations with the asthma phenotype, highlighting the recently proposed roles these cell types play in structuring EIB[21,22]. Finally, even where exercise regulates gene expression in ostensibly "disease-like" directions, it may be that many phenotypes as those above manifest when inflammatory, hunger-regulating, or other effects of exercise occur without having first been induced by exercise. We hypothesize that by subjecting the body to disease-like stresses, regular exercise elicits adaptation to the symptoms of those diseases, reducing the risk of their manifestation from the disease itself. In this light, the presence of their signal here may also be expected.

At the gene level, several of the highlighted genes in the DEG / PrediXcan intersection were supported by prior literature. In the cholesterol phenotype, *FADS2*[23], *PNKD*[24], and *OGDH*[25], *TMBIM1*[26], *APOB*[27], and ABCG8[28] have all been implicated previously, while *NDUFA13* and *AAMP* have not. For asthma and / or reduced lung function, links have been drawn to *BAG6*[29], CCNF[30], and *CRAT*[31], though other genes are mentioned in similar contexts to that explored in this work, also relying on integration of eQTL and GWAS association mapping (e.g., *FAM89B*[32]).

A notable limitation of this study may be that, despite their well-established use as an exercise model, rats are separated from humans by nearly 140 million years of evolution[33]. Comparison of exercise-independent age and sex effects, meanwhile, may be limited by differences in age between individuals in GTEx and MoTrPAC, as most humans in the GTEx v8 dataset were aged 50+[34,35] while trained F344 rats were uniformly under eight months of age and therefore well under the age of onset of the F344 rat equivalent of sex-specific, aging-related changes such as menopause[36]. These results may also have limited portability to non-European populations, as the GTEx sample comprises mostly European-descendant individuals. Identification of

rat-human gene orthology is another difficult problem, and important biology almost certainly lies within disease and exercise-responsive genes across species whose correspondence can not be easily established. But while species differences can complicate interpretation of exercise-induced regulation of orthologous genes, these models remain crucial and provide high levels of experimental compliance and tissue accessibility from individuals who are far more straightforward to motivate. As such, a unique aspect of the MoTrPAC rat exercise training data includes the availability of differential expression data across 15 distinct tissues, many impossible or impractical to collect in humans as part of an exercise study. Accelerated rat life history also makes it feasible to conduct experiments on exercise training adaptation on timescales relevant to their lifespan. It's simpler to regulate rat behavior than human behavior, reducing biases linked to non-compliance and attrition.

We expect future studies can benefit and expand on this work in several ways. Qualitative sex-specificity, a notable hallmark of exercise adaptation in humans[37,38], fell outside the scope considered here, though is afforded closer treatment in companion publications[39]. Future causal inferential work may use the genetic correlates of physical activity[40] as instruments to infer tissue-specific drivers of phenotypic adaptation[41] in humans. But analysis of experimental data from animal models will complement these efforts where genetic effects are weak (Fig. 2a), targeting causality directly to identify how tissue and organ systems adapt to exercise and influence a large variety of human traits and diseases. Finally, we expect that future studies may benefit from our work by evaluating specific loci therein for GxE interactions within large-scale human population biobanks. Combined, MoTrPAC's EET study provides a large-scale, cross-tissue map of changes in exercise adaptation that enables generating new mechanistic hypotheses on the disease impacts of exercise training.

## Methods
This study did not generate novel data, instead relying on data published in previous or concurrent studies. Animal procedures from the concurrent MoTrPAC PASS1B study[4] were approved by the University of Iowa's Institutional Animal Care and Use Committee.

### MoTrPAC EET study design
The MoTrPAC[42] Endurance Exercise Training Study is described in detail in the landscape manuscript[4] (data accessible at https://motrpac-data.org/data-access). In brief, both female and male F344 rats were subjected to treadmill exercise training, with tissues harvested at 1, 2, 4, and 8 weeks of training. All samples were taken 48 h after the last exercise bout, with the 8-week time point taken to correspond to the adapted state. In this work, we leverage data from a total of 738 extracted samples across 15 tissues and 47–50 rats per tissue that were subjected to RNA-sequencing and differential expression analysis.

### Differential expression analysis
Differential expression analysis (DEA) is described in detail by the main MoTrPAC manuscript[4]. Briefly, DEA was performed separately in each sex and tissue using filtered raw counts as input for `DESeq2`[43]. Likelihood ratio tests (*DESeq2*::`nbinomLRT()`) were used to identify genes that changed over the training time course in at least one sex while accounting for RNA-Seq technical covariates (RNA integrity number, median 5'-3' bias, percent of reads mapping to globin, and percent of PCR duplicates as quantified with Unique Molecular Identifiers). For each gene, male- and female-specific *p*-values were combined using the Fisher's *sum of logs* method. These meta-analytic *p*-values were adjusted across all RNA-Seq datasets using Independent Hypothesis Weighting (IHW) with tissue as a covariate[44]. Training-differential genes were selected at 5% IHW $\alpha$. Given the regression model of each gene described above, contrasts were made between each training timepoint (i.e., 1, 2, 4, or 8 weeks) and the sex-matched sedentary controls using *DESeq2*::`DESeq()` to calculate time- and sex-specific summary statistics.

### Correlation of differential analysis results
The nominal *p*-values and log fold-changes from the time- and sex-specific differential expression analysis results were transformed into standard normal random variables using `qnorm(p-value / 2, lower.tail = F) * sign(`$\log_2$`FC)` in base-R. These "z-scores" were organized into a gene-by-condition matrix, where conditions were tissue, sex, and timepoint combinations. The z-score matrix was filtered to include the set of genes that had no missing values across all conditions. We calculated the Spearman correlation between all pairs of conditions to quantify the concordance of the training effect across conditions.

### Graphical clustering of differential analysis results
Graphical clustering of differential analysis results is described in detail in the main MoTrPAC EET study manuscript[4]. All training-differential features at 5% IHW $\alpha$ were clustered into homogeneous patterns using their time- and sex-specific differential analysis z-scores. The statistical details are provided elsewhere[4,45–47]. Briefly, the expectation-maximization (EM) process of the *repfdr* algorithm was used to assign one of three simplified states to each z-score: −1 for down-regulation, 0 for null (no change), or 1 for up-regulation[45]. For each feature and timepoint, the simplified states from each sex were combined into one of nine possible states (−1, 0, or 1 for each sex). For example, the state "F1_M1" represented a feature that was up-regulated in both females (F1) and males (M1) at a given timepoint. Here, to focus on genes with sex-consistent training effects in the trained state, we selected genes that were assigned to the F1_M1 state (up-regulated in both sexes) or the F-1_M-1 state (down-regulated in both sexes) at 8 weeks. To enable comparison between genes expressed in rats and humans, we compiled a MoTrPAC rat-to-human ortholog map from GENCODE and RGD resources[4,48,49]. The distribution of those genes able to be matched to human orthologs across tissues is summarized in Fig. 1b.

### Open targets intersection
The Open Targets[50] database (Release 22.04) was downloaded on June 8th, 2022. Entries in this database represent curated sets of human genes with disease relationships established from multiple sources of evidence. We used the R-package *sparklyr*[51] to cross-reference differentially expressed rat genes to all orthologous Open Targets gene-trait direct associations at different evidence-score thresholds. The abundance of these associations were quantified on a tissue-specific and tissue-shared bases, comprising genes differentially expressed in three or more tissues. A table listing all genes, top trait associations, and corresponding tissues is provided in the Supplementary Files folder of the GitHub repo.

### Heritability analyses
We retrieved summary statistics (sumstats) for 114 published GWAS[5]. Using the program LDSC[9], we estimated SNP-heritability ($h^2_{SNP}$) for each GWAS in LDSC[10], including the default baseline annotation of 53 functional categories. We further estimated $h^2_{SNP}$ using MESC[11], and with the provided expression scores meta-analyzed over 48 GTEx tissues, estimated expression-mediated heritability ($h^2_{mediated}$) for our 114 traits, as well as the ratio of $h^2_{mediated}/h^2_{SNP}$.

LDSC was used to estimate overall proportion of and enrichment in $h^2_{SNP}$ across loci within a 100kb window of all sex-consistent 8w DE gene sets in each tissue following the "Cell type specific analyses" tutorial. We included here the baseline annotation, as well as an annotation comprising loci within 100kb of all expressed genes in each tissue. Finally, to assess the sensitivity of tissue-specific results on overlaps in gene sets between tissues, we estimated heritability and

heritability enrichment conditional on annotations corresponding to all other tissues alongside the baseline annotation.

## Human expression data & effect standardization

To assess the degree that exercise effects could overcome genetic and phenotypic variability of gene expression in a tissue, we used the GTEx database (version 8)[34]. To allow for a common scale between exercise DE and measures of gene expression in GTEx, we modified the GTEx pipeline to use a pseudolog ($\log_2(x+1)$) transformation in place of its default inverse-normal transform, otherwise keeping later steps in the pipeline intact. Next, we took the outputted expression matrices and residualized out the provided covariates using the `lm()` function in base-R (sex, the top 5 genotyping principal components, Sequencing platform, Sequencing protocol, and the suggested number of PEER factors in the GTEx documentation). On a per-gene basis, we then computed sample variances for each gene in each tissue, pooled across sex to reflect the sex-independent nature of exercise-induced DE. To regularize outlying variance estimates due to sampling effects, we fit an inverse-gamma distribution to tissue-specific sample variances using a maximum goodness-of-fit estimator implemented in the R-package *fitdistrplus*[52] by the function `fitdist()`. As the inverse-gamma is the conjugate prior of the variance term of a normal distribution with known mean, we adopted an Empirical Bayesian strategy to produce posterior estimates of each gene's expression variance. To allow for heterogeneity in this term across sex, we did this separately for male- and female-coded individuals in the GTEx study population. Additional details are provided in the Supplementary Methods. Across tissues, these empirical priors are plotted in the denominator of Fig. 2a. For each gene, we then took the square root of the posterior mean of inferred $\log_2$expression variance ($\sqrt{\text{Var}(\log_2(\text{gene expression}))}$) to estimate within-population standard deviation of the magnitude of gene expression. We then divided estimated exercise DE by these values to produce standardized estimates in units of within-tissue phenotypic standard deviation ($\text{SD}_{\text{pheno}}$). Further, these estimates were conditioned on both sex and population (quantile plots in upper panels of Fig. 2a).

To estimate the scale of genetic influence on gene expression, we used the software Plink[53] and GCTA[54], specifically GCTA-GRM[55], to estimate $h^2_{\text{SNP}}$ of each gene's expression, using the same covariates as before. In contrast to prior work estimating $h^2_{\text{SNP}}$ in GTEx' inverse-normal transformed gene expression matrices[56], we focused on obtaining estimates on a gene-specific basis, and so constrained output to be bounded between 0 and 1.

We then took these estimates, which represent the proportion of expression variance able to be explained by linear effects at the SNP level, and multiplied them by the estimates of expression variance, dividing our estimates of exercise-induced DE by the square root of that product to obtain exercise-effect sizes in units of genetic (SNP) standard deviation ($\text{SD}_{\text{geno}}$). Many of these $h^2_{\text{SNP}}$ point estimates were at or near 0, resulting in extreme standardized effect sizes. As a further filter, we thresholded on significance (IHW $\alpha = 0.10$, with tissue as a covariate) to focus on confidently heritable genetically regulated expression. This removed ≈92% of gene x tissue pairs (583,238/632,738), leaving 49,500 for later analysis and use in figures.

## Cross-referencing exercise-training genes with human TWAS

To identify specific genes where exercise-training effects may have the potential to mediate traits, we cross-referenced exercise-genes against transcriptome-wide association results (TWAS). Specifically, we downloaded S-PrediXcan[13] output[5] for 114 GWAS and MASHR-based expression models using GTEx v8, filtering by significance (IHW $\alpha = 0.05$, with tissue x trait pairs as a covariate), and intersected with genes that were differentially expressed due to exercise at 8W in a sex-consistent manner, i.e., members of the nodes "8w_F1_M1" and "8w_F-1_M-1". 99 of 114 traits had a nonzero intersect in at least one tissue.

To assess potential enrichments in these intersections, we compared the observed count of S-PrediXcan hits in the DEG sets against those outside the DEG sets, adopting a tractable Binomial approximation to the Bernoulli distribution to test for enrichment or depletion of genes under a multilevel Bayesian model, following ref. 57. This approach allowed us to partially pool information across tissues and traits, avoiding the need for post-hoc multiplicity adjustment[58], as multiplicity is explicitly built into the inference model itself through flexible regularization of model parameters towards 0. Specifically, we fit a model of the form:

$$y_{i,j}^{\text{DEG}} \sim \text{Binomial}(n_{i,j}^{\text{DEG}}, f(\pi_{i,j}^{\text{DEG}})) \tag{1.1}$$

$$y_{i,j}^{\neg\text{DEG}} \sim \text{Binomial}(n_{i,j}^{\neg\text{DEG}}, f(\pi_{i,j}^{\neg\text{DEG}})) \tag{1.2}$$

$$\pi_{i,j}^{\text{DEG}} = \pi_{i,j} + \frac{\alpha + \beta_i + \gamma_j + \epsilon_{i,j}}{2} \tag{1.3}$$

$$\pi_{i,j}^{\neg\text{DEG}} = \pi_{i,j} - \frac{\alpha + \beta_i + \gamma_j + \epsilon_{i,j}}{2} \tag{1.4}$$

$$\alpha \sim \text{Normal}(0, 1) \tag{1.5}$$

$$\beta_i \sim \text{Multi-Normal}(\vec{\mathbf{0}}, \sigma_\beta^2 \Sigma_i) \tag{1.6}$$

$$\gamma_j \sim \text{Multi-Normal}(\vec{\boldsymbol{\mu}}_k, \sigma_\gamma^2 \Sigma_j) \tag{1.7}$$

$$\epsilon_{i,j} \sim \text{Multi-Normal}(\vec{\mathbf{0}}, \sigma_\epsilon \Sigma_{i \times j}) \tag{1.8}$$

$$\mu_k \sim \text{Normal}(0, \sigma_\mu) \tag{1.9}$$

$$\pi_{i,j} \sim \text{Multi-Normal}(\eta_j, \sigma_\pi \Sigma_{i \times j}) \tag{1.10}$$

$$\eta_j \sim \text{Multi-Normal}(\vec{\boldsymbol{\lambda}}_k, \sigma_\eta^2 \Sigma_j) \tag{1.11}$$

$$\lambda_k \sim \text{Normal}(\mu, \sigma_\lambda) \tag{1.12}$$

$$\mu \sim \text{Normal}(0, 2) \tag{1.13}$$

$$\sigma_{\beta,\gamma,\epsilon,\mu,\pi,\eta,\lambda} \sim \text{Half-Normal}(0, 1) \tag{1.14}$$

Notation for this model is summarized in Supplementary Table 2, but in brief: the intersect size $y_{i,j}^{\text{DEG}}$ in tissue $i \in \{1, 2, ..., 15\}$ and trait $j \in \{1, 2, ..., 99\}$ was binomially distributed, with $n_{i,j}^{\text{DEG}}$ giving the total number of genes in that tissue that were differentially expressed at 8W and expressed at any level in the PrediXcan analysis (i.e., disregarding genes that were not expressed in both samples). The function $f()$ can be any function mapping $\mathbb{R} \to (0,1)$, but here was the inverse-logit function. On the logit-scale, $\pi_{i,j}^{\text{DEG}}$ was expressed as a deviation from a mean $\pi_{i,j}$, with an equal and opposite deviation to the log-odds of observing a PrediXcan hit in the complementary set, defined as all expressed genes that were not differentially expressed at 8W in a sex-consistent manner. This deviation term had four components: a tissue difference $\beta_i$, a trait difference $\gamma_j$, a tissue x trait difference $\epsilon_{i,j}$, and an overall difference $\alpha$. Adding and subtracting half from $\pi_{i,j}$ to produce $\pi_{i,j}^{\text{DEG}}$ and $\pi_{i,j}^{\neg\text{DEG}}$, respectively, was done to prevent specifying greater prior uncertainty on one of the two composite probability parameters.

The various scale parameters, $\sigma$, served to adaptively regularize estimates of each difference term towards their mean. Otherwise, we nested trait difference effects $\gamma_j$ in trait category difference effects $\mu_k$, where $k \in \{1, 2, \ldots, 12\}$ indexes previously designated trait categories[5], i.e., members of the set {Psychiatric, Aging, Cardiometabolic, Allergy, Digestive, Immune, Endocrine, Skeletal, Anthropometric, Hair, Blood, Cancer}. If traits in a particular category showed consistent evidence of deviation, partial pooling shrunk estimates towards their respective *mean* hyperparameters, allowing them to share information to the extent the model could detect information to be shared. We use a similar model structure to express the overall location parameter, $\pi_{i,j}$.

Pseudo-replication across tissues and traits amplifies signals that inform higher-level parameters, leading the inference model to mistake interdependent effects as independent evidence for enrichment. When aggregating many Bernoulli random variables to a single binomial, signals of gene interdependence that would otherwise prevent this are lost. To address this, we introduced parameters $\Sigma_i$, $\Sigma_j$, and $\Sigma_{i \times j}$, corresponding to $i \times i$ tissue, $j \times j$ trait, and $(i \cdot j) \times (i \cdot j)$ tissue $\times$ trait correlation matrices, respectively. For tractability, we then fixed these to maximum-likelihood estimates of each respective gene-wise correlation matrix under a bivariate probit, which we fit marginally across all DEGs, PrediXcan hits (jointly across tissues), and DEG $\times$ PrediXcan intersects using the *nlm* (non-linear minimization) algorithm[59] implemented in and accessed through the R-packages *stats* and *optimx*[60]. As we fit each pairwise correlation individually, rather than simultaneously, there was no guarantee that the resulting correlation matrix is positive semi-definite. To ensure this constraint is met and all pairwise correlations are jointly possible, we transformed the pairwise-estimated correlation matrices with Higham's algorithm[61] implemented in the R-package *Matrix*[62] function `nearPD()` before proceeding further.

We fit this model in Stan[63] using *CmdStanR*[64] and in Fig. 4e–g visualize marginal posterior distributions for tissue, trait, and trait category difference effects as violin plots using the R-package *vioplot*[65]. Additionally, where the composite difference effect for a particular cell in Fig. 4a finds > 95% of its posterior mass to one side of 0, we colored its upper or lower corner with a red or blue triangle to signify enrichment or depletion of that tissue x trait combination, respectively. To accommodate this and subsequent models' challenging posterior geometry, we used a non-centered parameterization, running four separate and randomly initialized chains for $2.5 \times 10^3$ warmup and $2.5 \times 10^3$ sampling iterations, with a target acceptance rate $\delta$ of 0.95. To diagnose pathologies in the MCMC output and confirm adequate convergence, mixing, and sampling intensity, we used the *posterior* package[66] for MCMC diagnostics, requiring that all model parameters, as well as posterior density and likelihood, receive $\hat{r} < 1.01$ and both bulk and tail Effective Sample Size (ESS) > 500, in addition to requiring that < 0.05% of iterations end in a divergence.

To complement this analysis, we also performed frequentist Gene Set Enrichment Analysis using the function `fgsea()` implemented in the R-package *fgsea*[67]. Specifically, for each trait $\times$ tissue pair, we assessed enrichment of the set of DEGs in that tissue in the list of -$\log_{10}(p$-values) of mutually expressed, orthologous genes' $p$-values from PrediXcan, applying a Bonferroni correction to the output. To aggregate across these interdependent tests and assess tissue- and trait-level enrichment, we took the harmonic mean of subtest $p$-values corresponding to each grouping[68], applying a similar FWER adjustment to the output ($\alpha = 0.05$). We leveraged the same meta-analytic procedure over trait-level enrichments to aggregate to trait categories. Finally, we explored an alternative approach to aggregating multi-tissue GSEA within traits. As a more stringent set of multi-tissue responsive genes, we took the set of all genes differentially expressed in three or more tissues. For PrediXcan $p$-values, we took the harmonic mean of PrediXcan $p$-values for each gene across all studied tissues prior to -$\log_{10}$ transformation. These were input into conventional

GSEA, and output from all of the above comparisons was visualized in Supplementary Fig. 3.

## Proportion of disease-like effects

To assess the proportion of DE acting in disease-like directions relative to each phenotype (the product of the direction of DE and the direct of association from PrediXcan), we applied another Bayesian multilevel model, comparing the observed, unweighted frequency of positive effects against a "null" frequency of 0.5 (equivalently, against log-odds of 0):

$$y_{i,j} \sim \text{Binomial}\left(n_{i,j}, f(\pi_{i,j})\right) \tag{2.1}$$

$$\pi_{i,j} \sim \text{Normal}\left(\vec{\boldsymbol{\mu}}_j, \sigma_j\right) \tag{2.2}$$

$$\vec{\boldsymbol{\mu}}_j \sim \text{Multi-Normal}\left(\vec{\boldsymbol{0}}, \text{SRS}^\text{T}\right) \tag{2.3}$$

$$R = G_{\text{SNP}} \times \theta + I \times (1 - \theta) \tag{2.4}$$

$$\theta \sim \text{Beta}\,(1, 1) \tag{2.5}$$

$$\text{diag}(S) = \delta e^{\gamma_k} \tag{2.6}$$

$$\gamma_k \sim \text{Normal}\,(0, \sigma_\gamma) \tag{2.7}$$

$$\sigma_j = \rho e^{\lambda_j} \tag{2.8}$$

$$\lambda_j \sim \text{Normal}\,(0, \sigma_\lambda) \tag{2.9}$$

$$\rho, \delta \sim \text{Half-Normal}\,(0, 2) \tag{2.10}$$

$$\sigma_{\gamma,\lambda} \sim \text{Half-Normal}\,(0, 1) \tag{2.11}$$

Unlike for their overall frequency, signal for the directionality of effect (more or less disease-like) cannot be shared across traits or within trait categories, as the traits themselves vary in whether they are harmful, neutral, or beneficial. Instead, we perform partial pooling across the scale of these differences, across both trait categories and within traits themselves. Notation for this model is summarized in Supplementary Table 3, but in brief: we estimate overall scale parameters ($\rho$, $\delta$), and then estimate a log-normally distributed multiplicative factor ($\lambda_j$, $\gamma_k$) to scale each on a trait-wise and trait-category-wise basis, respectively. To accommodate per-trait interdependence in the direction of deviation, we invert Cheverud's conjecture[69], using our previously estimated $G_{\text{SNP}}$ as a proxy for an environmental correlation matrix (Supplementary Fig. 1a). As these genetic correlations were estimated pairwise, positive-semidefiniteness (PSD) of the whole correlation matrix is not guaranteed. To satisfy the PSD constraint of $G_{\text{SNP}}$, we substituted the nearest PSD correlation matrix output from Higham's algorithm[61], implemented in the R-package *Matrix*[62] function `nearPD()`. To allow flexibility in this modeling assumption, we compute a linearly weighted average of this matrix and the identity matrix (i), estimating the weight parameter $\theta$ from a flat Beta prior. MCMC sampling parameters were specified and diagnostics performed as previously described.

We examined the two non-anthropometric traits with the highest posterior means in Fig. 6, tracing the proportion of effects of the 8W gene set backwards to the first week. Where individual genes are not

assigned to a graphical node signifying differential expression, their contribution to the count of positive effects was taken to be 0.5 when calculating the overall proportion. We include in these tissue-specific trajectories the set of genes corresponding to each tissue, their direction of effect on the trait, and their standardized effect size from Fig. 2b. Similar figures to Fig. 6 for all other traits may be found in the GitHub repository mentioned below.

## Reporting summary

Further information on research design is available in the Nature Portfolio Reporting Summary linked to this article.

## Data availability

This study did not generate novel data, relying instead on previously or concurrently published data. MoTrPAC PASS1B data (https://doi.org/10.1101/2022.09.21.508770) used here have been deposited at https://motrpac-data.org/data-access. Inquiries regarding access to these data should be sent to motrpac-helpdesk@lists.stanford.edu. Further resources are available at motrpac.org and motrpac-data.org. Where it would be difficult to re-host large datasets from GTEx[34], Open Targets[50], and PrediXcan[5], we provide download links in the documentation of the associated code repository. Source data to generate all figures seen here are provided with this paper in the form of *.RData objects. These contain all necessary processed data to fully and quickly reproduce all paper figures using the scripts contained in https://github.com/NikVetr/MoTrPAC_Complex_Traits/tree/main/scripts/figures. Source data are provided with this paper.

## Code availability

We provide end-to-end scripts to perform all analyses described above in a GitHub repository[70] located at the following URL: https://github.com/NikVetr/MoTrPAC_Complex_Traits. Additionally, we provide scripts to generate all figures, as well as intermediate data files corresponding to compiled results at each level of analysis (MCMC output, Open Targets associations, cross-referenced DEG-PrediXcan intersects, aggregated GCTA output, and relative effect sizes).

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

## Acknowledgements

We would like to thank Michael Gloudemans, Daniel Nachun, Bob Carpenter, Andrew Gelman, Laurens van de Wiel, Andrew Marderstein, Bruna Balliu, Kim Huffman, and Kate Gates for their valuable input on many parts of the analyses presented above. We would also like to thank Marty Walsh, John Williams, Matt Wheeler, and other members of MoTrPAC for their crucial feedback on this work. Finally, we would like to acknowledge the entire MoTrPAC team, including PASS, CAS and BIC, for their indispensable contributions in generating the exercise-response data used here. MoTrPAC is supported by NIH grants U24OD026629 (MSG, Bioinformatics Center), U24DK112349 (MSG), U24DK112342 (MSG), U24DK112340 (MSG), U24DK112341 (MSG), U24DK112326 (MSG), U24DK112331 (MSG), U24DK112348 (SBM, Chemical Analysis Sites), U01AR071133 (MSG), U01AR071130 (MSG), U01AR071124 (MSG), U01AR071128 (MSG), U01AR071150 (MSG), U01AR071160 (MSG), U01AR071158 (MSG, Clinical Centers), U24AR071113 (MSG, Consortium Coordinating Center), U01AG055133 (MSG), U01AG055137 (MSG), and U01AG055135 (MSG, PASS/Animal Sites). Research reported in this publication was supported by the National Library of Medicine of the National Institutes of Health under award number T15LM007033 (N.G.V.).

## Author contributions

NGV, NRG, and SBM collectively conceived of and designed the analysis strategy underlying this work. NGV implemented most analysis and figure code, with NRG facilitating access and advising processing of GTEx, GWAS, and MoTrPAC data. NGV and NRG performed testing and validation, as well as compiling online materials. NGV drafted the manuscript, which then received extensive edits and suggestions from NRG and SBM. MSG provided substantive feedback and advice

throughout all parts of this work. All authors approved the manuscript prior to submission.

## Competing interests

S.B.M. is a consultant for BioMarin, MyOme and Tenaya Therapeutics. These companies are broadly interested in treatments for rare and common genetic diseases but had no input on any component of this study. The authors have no other competing interests to declare.

## Additional information

## MoTrPAC Study Group

Joshua N. Adkins[2], Brent G. Albertson[3], David Amar[1], Mary Anne S. Amper[4], Jose Juan Almagro Armenteros[1], Euan Ashley[1], Julian Avila-Pacheco[5], Dam Bae[6], Ali Tugrul Balci[4], Marcas Bamman[7], Nasim Bararpour[1], Elisabeth R. Barton[8], Pierre M. Jean Beltran[5], Bryan C. Bergman[9], Daniel H. Bessesen[9], Sue C. Bodine[6], Frank W. Booth[10], Brian Bouverat[8], Thomas W. Buford[7], Charles F. Burant[11], Tiziana Caputo[3], Steven Carr[5], Toby L. Chambers[12], Clarisa Chavez[1], Maria Chikina[4], Roxanne Chiu[1], Michael Cicha[6], Clary B. Clish[5], Paul M. Coen[13], Dan Cooper[14], Elaine Cornell[15], Gary Cutter[7], Karen P. Dalton[1], Surendra Dasari[16], Courtney Dennis[5], Karyn Esser[8], Charles R. Evans[11], Roger Farrar[8], Facundo M. Fernández[17], Kishore Gadde[18], Nicole Gagne[15], David A. Gaul[17], Nicole R. Gay ®[1], Yongchao Ge[4], Robert E. Gerszten[5], Bret H. Goodpaster[13], Laurie J. Goodyear[3], Marina A. Gritsenko[2], Kristy Guevara[4], Fadia Haddad[14], Joshua R. Hansen[2], Melissa Harris[18], Trevor Hastie[1], Krista M. Hennig[1], Steven G. Hershman[1], Andrea Hevener[19], Michael F. Hirshman[3], Zhenxin Hou[20], Fang-Chi Hsu[21], Kim M. Huffman[22], Chia-Jui Hung[1], Chelsea Hutchinson-Bunch[2], Anna A. Ivanova[20], Bailey E. Jackson[6], Catherine M. Jankowski[9], David Jimenez-Morales[1], Christopher A. Jin[1], Neil M. Johannsen[18], Robert L. Newton Jr[18], Maureen T. Kachman[11], Benjamin G. Ke[23], Hasmik Keshishian[5], Wendy M. Kohrt[9], Kyle S. Kramer[6], William E. Kraus[22], Ian Lanza[16], Christiaan Leeuwenburgh[8], Sarah J. Lessard[3], Bridget Lester[12], Jun Z. Li[11], Malene E. Lindholm[1], Ana K. Lira[6], Xueyun Liu[20], Ching-ju Lu[8], Nathan S. Makarewicz[3], Kristal M. Maner-Smith[20], D. R. Mani[5], Gina M. Many[2], Nada Marjanovic[4], Andrea Marshall[6], Shruti Marwaha[1], Sandy May[15], Edward L. Melanson[9], Michael E. Miller[21], Matthew E. Monroe[2], Stephen B. Montgomery ®[1] ✉, Samuel G. Moore[17], Ronald J. Moore[2], Kerrie L. Moreau[9], Charles C. Mundorff[5], Nicolas Musi[24], Daniel Nachun[1], Venugopalan D. Nair[4], K. Sreekumaran Nair[16], Michael D. Nestor[2], Barbara Nicklas[21], Pasquale Nigro[3], German Nudelman[4], Eric A. Ortlund[20], Marco Pahor[8], Cadence Pearce[5], Vladislav A. Petyuk[2], Paul D. Piehowski[2], Hanna Pincas[4], Scott Powers[8], David M. Presby[3], Wei-Jun Qian[2], Shlomit Radom-Aizik[14], Archana Natarajan Raja[1], Krithika Ramachandran[3], Megan E. Ramaker[22], Irene Ramos[4], Tuomo Rankinen[18], Alexander (Sasha) Raskind[11], Blake B. Rasmussen[25], Eric Ravussin[18], R. Scott Rector[10], W. Jack Rejeski[21], Collyn Z-T. Richards[6], Stas Rirak[4], Jeremy M. Robbins[5], Jessica L. Rooney[15], Aliza B. Rubenstein[4], Frederique Ruf-Zamojski[4], Scott Rushing[21], Tyler J. Sagendorf[2], Mihir Samdarshi[1], James A. Sanford[2], Evan M. Savage[17], Irene E. Schauer[9], Simon Schenk[6], Robert S. Schwartz[9], Stuart C. Sealfon[4], Nitish Seenarine[4], Kevin S. Smith[1], Gregory R. Smith[4], Michael P. Snyder[1], Tanu Soni[11], Luis Gustavo Oliveira De Sousa[6], Lauren M. Sparks[13], Alec Steep[11], Cynthia L. Stowe[21], Yifei Sun[4], Christopher Teng[1], Anna Thalacker-Mercer[7], John Thyfault[26], Rob Tibshirani[1], Russell Tracy[15], Scott Trappe[12], Todd A. Trappe[12], Karan Uppal[20], Sindhu Vangeti[4], Mital Vasoya[4], Nikolai G. Vetr ®[1] ✉, Elena Volpi[25], Alexandria Vornholt[4], Michael P. Walkup[21], Martin J. Walsh[4], Matthew T. Wheeler[1], John P. Williams[27], Si Wu[1], Ashley Xia[27], Zhen Yan[23], Xuechen Yu[4], Chongzhi Zang[23], Elena Zaslavsky[4], Navid Zebarjadi[1], Tiantian Zhang[20], Bingqing Zhao[1] & Jimmy Zhen[1]

[2]Pacific Northwest National Laboratory, Richland, WA, USA. [3]Joslin Diabetes Center, Boston, MA, USA. [4]Icahn School of Medicine at Mount Sinai, New York City, NY, USA. [5]Broad Institute, Inc, Boston, MA, USA. [6]University of Iowa, Iowa City, IA, USA. [7]The University of Alabama at Birmingham, Birmingham, AL, USA. [8]University of Florida, Gainesville, FL, USA. [9]University of Colorado Anschutz Medical Campus, Aurora, CO, USA. [10]University of Missouri, Columbia, MO, USA. [11]University of Michigan, Ann Arbor, MI, USA. [12]Ball State University, Muncie, IN, USA. [13]AdventHealth Translational Research Institute, Orlando, FL, USA. [14]University of California, Irvine, CA, USA. [15]University of Vermont, Burlington, VT, USA. [16]The Mayo Clinic, Rochester, MN, USA. [17]Georgia Institute of Technology, Atlanta, GA, USA. [18]Pennington Biomedical Research Center, Baton Rouge, LA, USA. [19]University of California, Los Angeles, CA, USA. [20]Emory University, Atlanta, GA, USA. [21]Wake Forest University, Winston-Salem, NC, USA. [22]Duke University, Durham, NC, USA. [23]University of Virginia School of Medicine, Charlottesville, VA, USA. [24]University of Texas Health Science Center, San Antonio, TX, USA. [25]University of Texas Medical Branch, Galveston, TX, USA. [26]University of Kansas Medical Center, Kansas City, KS, USA. [27]National Institutes of Health, Bethesda, MD, USA.

