## [Peer Review File · Nature Communications]

REVIEWER COMMENTS

Reviewer #1 (Remarks to the Author):

See attached pdf file

Reviewer #2 (Remarks to the Author):

General comments: Vetr et al present an analysis of gene expression from the MoTrPAC EET study, and attempt to relate gene expression differences to human disease using GWAS and GTEx. The study design is certainly novel, and the analytic methods are sophisticated and their approach to human disease is unbiased. However, the assumptions / premise are difficult to accept, some of their findings strain credibility, and the overall presentation is quite difficult to understand.

1. Premise: the validity of the premise should be tested.

a. The overall premise appears to be that genes that are differentially expressed when rats exercise can be linked to genes that are differentially expressed by trait- or disease-associated genetic variants in humans, and that these links can lead to genes that can modulate human disease. It seems that this premise needs to be established first. It is not obvious that non-genetic regulation of genes in the same direction to that induced by genetic susceptibility variants will have the same mechanism / effect. The etiology of the change in gene expression, as well as the timing, cell-type, and context, are all likely important, in fact, there are cases where the measured gene expression in disease is the opposite direction than that predicted by the genetic variant.

b. Tissues: It is an even further extrapolation to test across all tissue types and diseases, many of which likely have no relationship to exercise. Notably, GTEx is from older deceased donors, and known to have issues with expression - see for example McCall 2016 AJHG. The authors use heritability of gene expression which is population / cohort specific.

c. The conclusion of the paper appears to be that there is widespread enrichment (the point estimates all seem to be > 1) of differentially expressed genes from exercise in rats to PrediXcan genes from human GWAS across all tissues and traits (e.g. Figure 3e). Are the authors sure that the null distribution of overlap is correct? Both PrediXcan and DEG genes could have similar bias. Could a random pairing of tissue / PrediXcan results be used to define a null distribution and then look for relative enrichment?

d. MoTrPAC gene expression: the low concordance of differentially expressed genes across tissues - even those of skeletal muscle - is surprising. Is this due to cell type differences, scaling or normalization, or a result of some other issue? For example, it seems like LDLR might change in a similar direction in all

tissues even if the magnitude is different. This result is also confusing in light of the plot in the supplement which appears to show positive correlations between differential expression at different weeks between tissues? Perhaps, this information is in reference 4 (the 'main' manuscript?) which is not available to the reviewer.

e. The rationale for choosing Week 8 expression should be clarified. Would a model that maximized the data and correlation among samples be superior to identify exercise induced change, consider for example `duplicateCorrelation()` in limma to address the relatedness among samples.

2. Statistical methods:

a. Please further describe the calculation used for “these empirical priors”. The current description provides only an overview of the approach - the calculation may involve the sample variance of expression level, or a component from the method of moments?

b. The standardization process described at line 439-447 is confusing - is there information on the theory or reference? It is difficult to understand why fold changes are divided by the estimated standard deviation of gene expression level. This is an important concept in this article and needs to be specified.

Note for this section, Figure 1c and Figure 1d appear to be incorrectly referenced.

c. Please clarify the dimensions for defined vectors or matrices in Table S2 (With predefined number of genes and hits for i and j).

3. Presentation:

a. Clarity in text:

The analytic pathways and assumptions are complex. It is not clear from the text, for example, whether the author is discussing rat or human datasets. The figure does make things clearer, but the text should match. An effort should be made to increase readability, e.g. one of the figure captions reads, "...denominator lie a stacked histogram of h2 SNP estimates for expression scores from GTEx with a p-value > 0.05 after IHW correction, and the collection of inverse-gamma distributions serving to regularize log2 normalized expression scores after covariates had been residualized out."

b. Clarity in figures:

Too many similar colors that don't have meaning make it hard to interpret. For example, what is the message of Figure 1b (lower half)? Would consider simplifying: instead of trying to present all tissue types / GWAS / etc. consider limiting and / or grouping and using Tables to show all data.

The number of trait x gene pairs will be affected by the number of DEGs. This could potentially make it difficult to compare values across the lines.

4. Human traits and diseases: some of these conclusions seem highly suspect.

a. The apparently random selection of GWASs seems to result in increased noise without clear signal. There are results such as - 4 DEGs in the hippocampus related to body fat percentage and 9 colon genes related to asthma? It seems that expanding the list of metabolic traits would make more sense.

b. The discussion uses weak data and speculation. It is unclear how to relate human height effects which are likely relevant to childhood / growth, in adult rats that are exercising. Worse, to suggest that exercise causes gene expression that mimics and therefore worsens asthma based on splenic (and kidney and hippocampus - Figure 5) expression, and genes with weak evidence of genetic causality (references 54-57), is not credible and is a potentially dangerous public health message. Please modify.

c. The directionality is confusing: There are "positive" effects for BMI, asthma, and "negative" effects on height? This seems to suggest that exercise would increase BMI and body fat and asthma; and reduce height? Is this correct?

d. There are several confusing aspects of the chosen traits (why are there two asthma GWASs, and why is one Allergy the other as Autoimmune)? Why was pulmonary embolism, and father's age at death chosen?

Minor comments:

Figure 1d: "h² SNP estimates for expression scores from GTEx with a p-value > 0.05" so all non-significant heritabilities were included? The plot also seems to show that estimated h² SNP - most are between 80-100%?

Consider using exercise induced bronchoconstriction instead of exercise induced asthma.

Intracorneal volume (typo).

Reviewer #3 (Remarks to the Author):

SUMMARY:

This paper performed by Vetr et al. aims to use a series of bioinformatic analyses to identify the integration of exercise on gene regulation in different rat tissues and the impacts of exercise training on disease risk. The authors stepped down from differentially expressed genes across 19 tissues to single nucleotide polymorphism heritability enrichment analysis and then their discovering gene-tissue-trait changes in the exercised rats. This paper establishes important approaches to analyze genome-wide data from multi-center preclinical studies and translate to human diseases in the scope of exercise adaptation.

THREE MAJOR CONCERNS that must be sufficiently addressed:

1.Line 779. Please provide your meaning of the word "compliance" in line 779.

2. Justify with references your usage of a single time point at 48 hours post exercise? You should provide other time points. For example, PGC1alpha transcription peaks at zero hours post exercise and PGC1Alpha mRNA peaks at one-hour post-exercise, is still increased 6 hrs post exercise and unchanged 24 hours post exercise in a biopsy from the middle portion of the vastus lateralis muscle of each leg from 19–24-year-old, male subjects in a paper cited over 664 times (Pilegaard et al. J Physiol. 2003 Feb 1;546(Pt 3):851-8. doi: 10.1113/jphysiol.2002.034850.). (See general comments #4 in the next section)

3. Inconsistency from Abstract to main text. The authors should carefully examine and proofread how many tissues were included in the study and their descriptions are consistent across Abstract, figures, and main text (e.g., it was mentioned 15 tissues were analyzed in Abstract but described 19 tissues were used for analysis in Methods)

GENERAL COMMENTS:

1. The main purpose of the study appears to be to discover the impact of exercise on trait genetics. While integrating human disease traits and the alterations of gene expressions in rat exercise training, it is highly possible that some of the human disease-related genes may not be translatable from rat orthologs to human gene sequences. We suggest that the authors should expand a single sentence in the discussion for the potential lack of obsolescence in translating rat sequences to human sequences at the sites of transcription factor bindings.

2. It has been well-established that male/female differences have an essential role in driving exercise adaptation (J Physiol. 2023 Feb;601(3):419-434. Epigenetics 2019 Jun;14(6):523-535). The candidate genes used in the study were the genes that went up in both sexes. As some genes respond in opposite directions in males and females to endurance-type exercise, the criteria used in the study may lose sex-specific gene regulation after exercise training. If no further sex-specific analysis are to be done, please address sex differences in the discussion as a limitation of the current study.

3. The protective/anti-protective effects of exercise training can depend on whether exercise is aerobic or is resistance type of exercise. We suggest that the authors should identify the term exercise in the Abstract and Discussion by using either “endurance exercise” or “aerobic exercise” to help readers to specify the findings in the current study are from endurance exercise training.

4. While using the term “exercise response”, it should be noted that under the current study design for sample collections (48-hour after last training period), the results would be more related to “baseline gene expression after exercise training” or “gene expression in resting tissues after exercise training”, which in other words, it is a response to exercise adaptation, but not a response immediately after exercise. “Exercise adaptation” and “exercise response” may not be interchangeable since one refers to

the training effect and the other refers to response after a bout of exercise. This may explain the results why fewer alterations were found at 1-week and 2-week time-point, since the effects of exercise were recovered from the last bout of exercise and exercise adaptation had not yet fully developed. Some mRNAs only change during the exercise-bout itself. Others only change for a few hours after exercise. For example, human PGF1alpha does not change during exercise, but PGC-1alpha is increased 2 hrs post-exercise, and then is at resting levels at 10 hrs post exercise. This point mirrors major concern #2.

5. Data visualization: Since the scope of the study is to investigate the influence of exercise training in gene regulation with inter- and intra-tissue levels, we recommend either using colorblind-safe labeling or adjusting the choice of colors in the current figures. Some of the color themes used in the current figures may be difficult to distinguish and interfere the interpretation of results from readers.

For example,

Fig. 2: BLOOD, HIPPOC, and LIVER

Fig. 3a and Fig. 4: color theme for the X-axis

Fig. 4: HIPPOC, HYPOTH, SPLEEN, and HEART

SPECIFIC COMMENTS for Methods and Results

Methods

1. The source of the data and the methods from MoTrPAC EET study, which refers to Ref#4 cited in this paper, has not been published while reviewing this manuscript. We suggest that the authors should cite the most recent version of the manuscript on BioRxiv (<https://www.biorxiv.org/content/10.1101/2022.09.21.508770v2>) and the publicly accessible NIH/GitHub from MOTrPAC (<https://commonfund.nih.gov/moleculartransducers>) (<https://motrpc-data.org/data-access>) to help readers to understand MoTrPAC study design and track MoTrPAC related publications extensively.

2. Line 277-289. The authors briefly described the study design of MOTrPAC EET and the sample collections were from female and male F344 rats in a total number of 50 rats across 19 tissues. However, only 874 samples were extracted. Does this suggest that only 46 rats were in the study ($874 \text{ samples} \div 19 \text{ tissues} = 46 \text{ rats}$?) or there were some missing data from some rats or tissues? As Line 311-316 mentioned, rats under sedentary condition were included as no exercise control in this study and analyses, please address it in the study design section and clarify the number of animals in each condition, time-point, and sex.

Results

1. The main findings for Fig. 1D and 1E did not appear in the Results for Fig 1. (Line 87-167) and may confuse readers. Since the identifiers of Fig. 1D and Fig. 1E showed in the later part of the main text (Line 447 and 649) for the first time. The authors need to correct this.

2. Lines 145-167, Page 3. We encourage the authors to use a table to present the results described from Line145-167 to help readers having a better understanding.

3. Figure 3.

a. The overall resolution for Fig 3 is visibly lower than other figures in the manuscript.

b. Please make sure that the description in Line 198-199 is consistent with the heading of Figure 3a.

Dear anonymous reviewers,

Thank you for your valuable feedback on our manuscript titled “**The impact of exercise on gene regulation in association with complex trait genetics**”, submitted to *Nature Communications*. We appreciate the time and effort you have devoted to reviewing our work.

We have carefully considered the comments you provided. Below you will find our point-by-point revisions or responses to each of them. We believe these revisions have substantially improved both the quality and clarity of our paper. We first quote each **Comment** verbatim, and then we detail our **Response**, highlighting any differences in the revised text, analysis, figures, or other materials. To proceed directly to responses corresponding to each reviewer, you may click the following: **Reviewer 1**, **Reviewer 2**, **Reviewer 3**. To proceed directly to a summary of our unsolicited changes and closing remarks, you may click **here**.

Reviewer 1

Comment: (1) How does gene expression fluctuate post-conclusion of exercise in rats? Specifically, why was 48 hours after last exercise bout chosen for tissue harvesting? Are the timepoints reasonable correlates for exercise training adaptation in humans? The manuscript may benefit from an expanded discussion of these topics in the introduction or methods.

Response: Study design decisions for PASS were not a component of this specific companion paper and are discussed in greater depth in the (now provisionally accepted) landscape paper¹, as well as in the introductory publication² and the animal protocol (accessible at motr-pac.org/protocols.cfm). An important feature to note is that the 48h time point coincides with the final time point collected in MoTrPAC's acute exercise study (ongoing) and that external studies show gene expression returning to baseline 48h after exercises (Sue Bodine, personal correspondence).

■

Comment: (2) It appears the main MoTrPAC EET study manuscript that is being referenced is not published. Therefore, the details of the differential expression analysis, graphical clustering, and general study design, for example, are not available. These details are important to understanding the analyses presented in this manuscript.

Response: We apologize for the omission – an error in our citation software prevented appropriate citation of the MoTrPAC EET preprint. The corresponding reference has been updated to:

Group, M. S. *et al. Temporal dynamics of the multi-omic response to endurance exercise training across en.* Pages: 2022.09.21.508770 Section: New Results. Oct. 2022. <https://www.biorxiv.org/content/10.1101/2022.09.21.508770v3> (2023).

This paper is now provisionally accepted and an updated version can be provided to the reviewers on request (we didn't attach it here since there are multiple supporting documents).

■

Comment: (3) In line 361, I believe there is a typo in the reference to Figure 1a.

Response: The text has been changed to refer to the correct subfigure: “The distribution of

those genes able to be matched to human orthologs across tissues is summarized in Fig. 1b.”

■

Comment: (4) With respect to the Open Targets database query, how are these evidence-score thresholds defined and what do they indicate?

Response: Here, we rely on evidence scores computed by Open Targets, which combine multiple lines of evidence, including those from “genetic associations, somatic mutations, known drugs, pathways/ systems biology, RNA expression, text mining, [and] animal models”³. However, across the distribution of DE rat-human orthologs there was limited evidence for natural breakpoints. For example, there were no visually obvious discontinuities in any of the panels of 1c:

Nor anything especially distinct in a histogram of all unique DE genes with human orthologs:

Though it may be argued that a second mode around 0.8 is visible above, with a breakpoint in the region of 0.6. Our textual description could have subsequently chosen a different quintile threshold, but for compactness we chose the top quintile, rather than the top two, preferring to provide the more continuous view in Figure 1c. However, it is correct to note the arbitrariness of this threshold, so we have modified the line:

“To this point, we observed 370 high-scoring (Open Targets evidence score > 0.8) disease genes from 251 traits across our 15 surveyed tissues...”

to read:

“To this point, we observed 370 high-scoring (Open Targets evidence score > 0.8, an arbitrary threshold chosen to select gene x trait relationships with high levels of supporting evidence) disease genes from 251 traits across our 15 surveyed tissues...”

Additionally, we provide three sets of four further supplemental files compiling associations

observed above a 0.6, 0.4, and 0.2 threshold.

■

Comment: (5) The table listing all genes, top trait associations, and corresponding tissues from Open Targets is not provided in Supplemental Table 1 (line 377). I cannot seem to find this table anywhere in the supplement.

Response: We again apologize for the omission. In case we are unable to successfully transmit these files through the submissions portal, please find them located in the following directory of the GitHub repo: github.com/NikVetr/MoTrPAC_Complex_Traits/tree/main/supplemental_files. We have resubmitted these with this revision and provide a link to the location of the repo in the manuscript.

■

Comment: (6) In line 439, the empirical priors are indicated to be plotted in the denominator of Figure 1c, but this figure indicates Open Targets associations.

Response: Thank you for the correction. We have amended the corresponding line to read “... *Fig 1d.*”

■

Comment: (7) For the S-PrediXcan results that were downloaded, were these summary statistics all derived using the same statistical model for genetically-regulated gene expression? S-PrediXcan can use gene expression prediction models trained via different approaches (i.e., MASHR vs elastic net regression).

Response: The summary statistics for GReX (also spelled *GREx*) input into S-PrediCan in *Barbeira et al. 2021*⁴ were obtained from MASHR-based models, per the documentation here: “*S-PrediXcan was run for the 114 harmonized and imputed traits, on eQTL and sQTL mashr prediction models*”, ultimately traceable to recommendations here.

We have clarified the source of these estimates in-text, amending the line, “*Specifically, we downloaded S-PrediXcan output for 114 GWAS...*” to read “*Specifically, we downloaded S-PrediXcan output for 114 GWAS and MASHR-based expression models using GTEx v8 ...*”

■

Comment: (8) It would be informative to see what proportion of SNP-heritabilities you see captured by other gene sets of the same size as those DE due to exercise training. If you were to randomly permute your gene sets and repeat the heritability enrichment analysis, this may present an informative baseline comparison for the results presented.

Response: Our initial intuitions, which we'd formed through pilot experiments and private re-implementation of LDSC, was that these would be distributed around the proportion of SNP heritabilities captured by all tested genes with identifiable rat-human orthologs, as both coding regions and conserved genes (likely enriched among identifiable rat-human orthologs) are included in LDSC's baseline annotation.

But performing these analyses in simulation replicate, we observe something distinctly different: a very similar overall pattern of results to those we'd found in the observed gene sets (note, we concatenated output corresponding to five independent randomly shuffled gene sets, resulting in a five-fold difference in the scale of the vertical axis):

(for context, both the LDSC github repo and comments by Finucane at github.com/bulik/ldsc/issues/129 state that the “Partitioned Heritability: Cell-type group analysis” and “Cell type specific analyses” tutorials perform equivalent analyses, with only the form of the output and the efficiency of the computation differing. Our suspicion is that the “cell-type specific analysis” may not be fully conditioning on the baseline or other annotations)

Given these comparisons, we have added strongly tempering language to our discussion of the LDSC output. But we still consider it to be of general interest, and so have not removed it entirely. Specifically, we have now added the following lines to the corresponding section of

our results:

“However, we performed simulation experiments using randomly sampled gene-sets of equivalent size to our original tissue-specific genesets. These produced highly similar distributions of p-values to those observed for the empirical data. As such, these results (Fig. 3) should be interpreted less in the framework of null-hypothesis significance testing and more descriptively, as a relative ordering of estimated magnitudes of effect.”

Another rationale for keeping this analysis is that it is possible that exercise adaptation genes are randomly distributed with respect to the heritabilities of many diseases and traits but the actual observed ordering could still inform what traits are most likely to be impacted by this adaptation. However, our text change hopefully provides a significant enough qualification. We do think it would be of broad interest for a future study to investigate how $G \times E$ and E effects are distributed with respect to underlying G changes for polygenic traits.

■

Comment: (9) It may also be interesting to perform a gene-set enrichment analysis via GSEA Pre-ranked tool to test for enrichment of genes in the exercise-DE sets among genes ranked by $-\log_{10}(\text{S-PrediXcan p-value})$. This would indicate whether exercise-DE genes tended to fall at the top of the ranked list of genes by TWAS p-value. This would be a nice supplement to the enrichment analysis performed under your multilevel Bayesian model.

Response: We had explored a few more conventional frequentist approaches for 2×2 contingency table analysis, but found sample sizes to be too small at the tissue \times trait level to reach conventional significance thresholds after multiplicity adjustment. Following your suggestion, we find a similar pattern holds – effects are too small and signals too weak, so while plenty of traits reach nominal significance, the number of combinations tested ensures none of them survive Bonferroni-correction. The top of the list overlaps strongly with traits and tissues featuring the most confident positive effects in Fig. 3d,f, however:

```

> hist(gsea_output$adj.pval, breaks = 0:100/100)
> head(gsea_output[order(gsea_output$pval),c("trait_name ", "tissue", "pval", "adj.pval")], 20)
  trait_name tissue      pval adj.pval
1:   Red_Blood_Cell_Count BLOOD 6.035192e-05 0.09565779
0:     Breast_Cancer BLOOD 6.731776e-04 1.00000000
0:   Crohns_Disease_UKBS ADRNL 9.480305e-04 1.00000000
0:   Triglycerides_NMR LIVER 2.312396e-03 1.00000000
0:   LDL_Cholesterol KIDNEY 3.385329e-03 1.00000000
1:   BMI_Active_Inds SPLEEN 3.904027e-03 1.00000000
2:   Balding_Pattern_3_UKB ADRNL 4.424549e-03 1.00000000
2:   Fasting_Insulin HEART 6.283566e-03 1.00000000
0:   Inflammatory_Bowel_Disease SPLEEN 6.357927e-03 1.00000000
0:   Alzheimers_Disease COLON 6.506648e-03 1.00000000
0:   Irritable_Bowel_Syndrome_UKBS KIDNEY 7.398980e-03 1.00000000
0:   Waist_Circumference_EUR SMLINT 8.444430e-03 1.00000000
3:   HDL_Cholesterol SKM-VL 8.444430e-03 1.00000000
1:   Irritable_Bowel_Syndrome_UKBS HIPPOC 8.743341e-03 1.00000000
4:   Asthma_TAGC_EUR SPLEEN 8.743341e-03 1.00000000
1:   Waist_Circumference_EUR SPLEEN 9.341162e-03 1.00000000
3:   High_Cholesterol_UKBS HIPPOC 9.938984e-03 1.00000000
5:   HDL_Cholesterol SMLINT 1.098517e-02 1.00000000
6:   HDL_Cholesterol KIDNEY 1.098517e-02 1.00000000
2:   Systolic_Blood_Pressure BLOOD 1.098517e-02 1.00000000

```

Plots Packages Help Viewer Presentation

Mik I 7⁰ Zoom 4a Export - Publish

Histogram of gsea_output\$adj.pval

gsea_output\$adj.pval

Frequentist meta-analysis under conditions of unknown test interdependence can be done in many ways – one recently proposed involves taking the weighted harmonic mean p-value of subtests⁵. Performing these meta-analyses likewise did not identify significant results after multiple testing adjustment:

```
> library(harmonicmeanp)
> L <- nrow(gsea_output)
> tissue_ps <- sapply(split(gsea_output$pval, gsea_output$tissue), function(tissps){
+   (length(tissps)/L)/sum((1/L)/tissps)
+ })
> tissue_thresh <- qharmonicmeanp(0.05, L) * table(gsea_output$tissue)/L
> tissue_thresh <- tissue_thresh[names(tissue_ps)]
> any(tissue_ps < tissue_thresh)
[1] FALSE
> head(sort(tissue_ps), n = 10)
      BLOOD      ADRNL      SPLEEN      LIVER      KIDNEY      LUNG      SMLINT
0.005785384 0.061172602 0.065940692 0.108702114 0.108944344 0.125767317 0.165163856
      HIPPOC      COLON      SKM-VL
0.165753071 0.167066034 0.185855144
>
> trait_ps <- sapply(split(gsea_output$pval, gsea_output$trait_name), function(traitps){
+   (length(traitps)/L)/sum((1/L)/traitps)
+ })
> trait_thresh <- qharmonicmeanp(0.05, L) * table(gsea_output$trait_name)/L
> trait_thresh <- trait_thresh[names(trait_ps)]
> any(trait_ps < trait_thresh)
[1] FALSE
> head(sort(trait_ps), n = 10)
      Red_Blood_Cell_Count      Breast_Cancer      Crohns_Disease_UKBS
              0.0008433503              0.0092154368              0.0126938357
      Triglycerides_NMR      HDL_Cholesterol      LDL_Cholesterol
              0.0245161791              0.0260089320              0.0322576259
      BMI_Active_Inds      Balding_Pattern_3_UKB      Irritable_Bowel_Syndrome_UKBS
              0.0373096935              0.0482161971              0.0499142326
      Waist_Circumference_EUR
              0.0520750714
```

Though unsurprisingly we again see similar patterns at this level of aggregation to those seen in Fig. 3d,f. Across these trait-specific p-values, we next aggregated across traits within trait categories, but did not find any significant results after multiple testing correction here, either:

```

> trait_category_ps <- sapply(split(trait_ps, trait_cat_key), function(traitcatps){
+   (length(traitcatps)/L_traits)/sum((1/L_traits)/traitcatps)
+ })
> trait_category_thresh <- qharmonicmeanp(0.05, L_traits) *
+   table(trait_cat_key) / L_traits
> trait_category_thresh <- trait_category_thresh[names(trait_category_ps)]
> any(trait_category_ps < trait_category_thresh)
[1] FALSE
> sort(trait_category_ps / (trait_category_thresh / 0.05))
trait_cat_key
      Blood      Cardiometabolic      Immune
0.1260528    0.5927232    0.9969603
      Cancer  Psychiatric-neurologic  Anthropometric
1.2944338    1.2982964    1.4051811
      Allergy  Hair morphology  Digestive system disease
4.5450511    6.1195011    7.8551419
Endocrine system  Aging  Skeletal system disease
15.8986490    19.3863772    22.6083292

```

We also explored an alternative way to aggregate GSEA within traits. For our target gene set, we took the set of all genes differentially expressed in three or more tissues as a more stringent set of multi-tissue responsive genes. For PrediXcan p-values, we took the harmonic mean of PrediXcan p-values for each gene across all studied tissues prior to transformation. But this approach also failed to produce any significant results at FWER $\alpha = 0.05$:

```

> hist(gsea_output_pooled_alt $adj.pval)
> head(gsea_output_pooled_alt [order(gsea_output_pooled_alt$pval),
  c("trait_name", "pval", "adj.pval")], 20)
  trait_name      pval adj.pval
1:   High_cholesterol_UKBS 0.000471686 0.05094209
0:         CHZDB_NMR 0.003422327 0.18480563
0:   Triglycerides_NMR 0.009042251 0.32552104
0:   LDL_cholesterol_NMR 0.013824839 0.34525475
0:   Triglycerides 0.022980004 0.34525475
0:   Migraine_UKB 0.026584663 0.34525475
1:   Migraine_UKBS 0.026973027 0.34525475
0:   LDL_cholesterol 0.027972028 0.34525475
1:   Neuroticism_UKB 0.029970030 0.34525475
2:   Birth_weight_UKB 0.031968032 0.34525475
1:   HDL_cholesterol_NMR 0.037962038 0.37271819
1:   Insomnia_UKB 0.058941059 0.53046953
0:   Heart_Rate 0.065934066 0.53946054
3:   Sleep_Duration 0.069930070 0.53946054
0: ER-positive_Breast_Cancer 0.080919081 0.55969031
2:   Epilepsy_UKB 0.082917083 0.55969031
1:   Alzheimers_Disease 0.101898102 0.61895999
1: Rheumatoid_Arthritis_UKBS 0.104895105 0.61895999
2:   Sleep_Duration_UKB 0.108891109 0.61895999
0:   Coronary_Artery_Disease 0.122877123 0.66353646

```

Packages Help Viewer Presentation

Publish

Histogram of gsea_output_pooled_alt\$adj.pval

gsea_output_pooled_alt\$adj.pval

Though again with broadly comparable output to the Bayesian model.

Despite not finding any results that hold at FWER $\alpha = 0.05$, we do agree that these analyses would provide a valuable complement to our original approach. Specifically, we compare the above output to that from the Bayesian model with a new supplemental figure, Figure S3:

Figure S3: Bayesian and frequentist methods broadly agree in evaluating DEG enrichment across PrediXcan output. Here, we compare enrichment results between posterior summaries from our multilevel enrichment model (horizontal axis, Fig. 4) and frequentist Gene Set Enrichment Analysis (GSEA). In a), we plot $-\log_{10}$ (Bonferroni-adjusted p-values) from GSEA of DEGs in the ranked list of unadjusted $-\log_{10}$ (p-values) from PrediXcan for each trait x tissue pair. In b), we aggregate across tissues by taking only genes that are differentially expressed in three or more tissues and performing GSEA across harmonic mean p-values across the 15 matched tissues from the PrediXcan output. In c-d), we aggregate output from a) by taking harmonic mean p-values across traits and tissues, respectively, and computing $-\log_{10}$ (FWER-adjusted p-values) for plotting. In e), we take the harmonic mean of unadjusted p-values from c) across trait categories and apply the same transformation as before.

Besides the above figure and figure caption, we have also included the following lines in the main text:

Methods:

“To complement this analysis, we also performed frequentist Gene Set Enrichment Analysis using the function `fgsea()` implemented in the R-package *fgsea*⁶. Specifically, for each trait \times tissue pair, we assessed enrichment of the set of DEGs in that tissue in the list of $-\log_{10}(\text{p-values})$ of mutually expressed, orthologous genes’ p-values from PrediXcan, applying a Bonferroni correction to the output. To aggregate across these interdependent tests and assess tissue- and trait-level enrichment, we took the harmonic mean of subtest p-values corresponding to each group-ing⁵, applying a similar FWER adjustment to the output ($\alpha = 0.05$). We leveraged the same meta-analytic procedure over trait-level enrichments to aggregate to trait categories. Finally, we explored an alternative approach to aggregating multi-tissue GSEA within traits. As a more stringent set of multi-tissue responsive genes, we took the set of all genes differentially expressed in three or more tissues. For PrediXcan p-values, we took the harmonic mean of PrediXcan p-values for each gene across all studied tissues prior to $-\log_{10}$ transformation. These were input into conventional GSEA, and output from all of the above comparisons was visualized in Fig. S3.”

Results:

“Conversely, none of the frequentist analyses of this overlap produced significant results at FWER $\alpha = 0.05$ ($-\log_{10}(0.05) \approx 1.30$), with the most significant result corresponding to the multi-tissue GSEA for *high cholesterol* at an adjusted p-value of ≈ 0.051 (Fig. S3, $ES \approx 0.63$, $\log_2 \text{err} \approx 0.48$). However, results were broadly concordant across the two approaches, and more confident posterior distributions corresponded to lower frequentist p-values, with intermediate positive Spearman’s ρ s for pairwise and trait-wise comparisons (Fig. S3a-c). Frequentist meta-analysis of tissue and trait-category enrichments were in less confident agreement, with the latter showing mild disagreement, though at $p \approx 0.53$ (output from `stats::cor.test` in R).”

Reviewer 2

Comment: (1a) The overall premise appears to be that genes that are differentially expressed when rats exercise can be linked to genes that are differentially expressed by trait- or disease-associated genetic variants in humans, and that these links can lead to genes that can modulate human disease. It seems that this premise needs to be established first. It is not obvious that non-genetic regulation of genes in the same direction to that induced by genetic susceptibility variants will have the same mechanism / effect. The etiology of the change in gene expression, as well as the timing, cell-type, and context, are all likely important, in fact, there are cases where the measured gene expression in disease is the opposite direction than that predicted by the genetic variant.

Response: This is a great point and one that we considered often when working on this paper (for example, protective homeostatic factors may be expressed in a disease context when measuring gene expression directly). This is precisely why we used genetic regulation of gene expression as our metric in human data instead of gene expression levels observed directly in disease. The expectation is that genetic regulation of gene expression, and subsequent colocalization with GWAS or TWAS provides information on mechanism (i.e. via gene expression) and also the risk and protective directions of gene expression changes that can't be obtained from looking at gene expression more directly in a disease context. This expectation is not unique to our study and is actually foundational to most colocalization and TWAS analyses in the literature. Most of our added rationale is that the loci that change in concordant (or discordant) directions to risk may override the underlying genetic factors mediated by expression. However, for the reasons you indicate, we can't further suggest beyond this rationale that exercise would be helpful or harmful in the observed context of a disease state. We also can't assume just from heritability analyses that the mechanisms are all through gene expression, hence our additional focus on TWAS.

The causal effects of interest here are those linking gene expression and trait variation, with genetic variation comprising "randomly" assigned instrumental variables that are themselves not influenced by neither the phenotype of interest nor gene expression nor any shared causes of either (ie, the exogeneity assumption) that block all backdoor paths between the two focal variables. As such, any other instruments could serve in their place, so long as they equally satisfy the instrumental variable assumptions (relevance, exogeneity, exclusion), and the use of multiple instruments (ie, multiple SNPs at multiple loci) helps with robustness against minor violations by intersecting patterns of shared causal effect. These causal effects may be

thought of as representing shared mechanism: in the case of several of our identified positive relationships between exercise and some pathology, the pathology may share similarities with the harmful effects of exercise to which one adapts, building resilience to them. When the acute exercise state is absent, benefits from the adaptation are felt. A trait like asthma may arise when one chronically stresses and inflames the lungs without having first exercised; pathologically high BF% may result when sedentary individuals eat with the appetites of the regularly exercising; high heart rate (enriched for intersects but not directional effects) is produced when one's heart beats at an exercising rate while not exercising. It's not that exercise adaptation elicits chronically *high heart rate* at rest, it's that transcriptional changes upstream and downstream from *high heart rate* are similar to those experienced by those regularly exercising.

We also strongly agree on the context-sensitivity of these results, and the extent to which these results do not incorporate sub-tissue cell type or temporal granularity. The causal effects estimated by TWAS are lifetime-averaged, where the expression \rightarrow trait relationship may only be most relevant during some critical developmental period. They are further limited in three more critical ways: 1) they represent variation independent of its instantiation in any particular individual, with some complex network of intermediate variables on the path between gene and trait expression. Hence relationships between gene expression and variables such as parents' age at death: discernible in the same way that parents' age at death is heritable, through the inheritance of latent health or behavioral factors, and not because of especially parricidal genes. Additionally, 2) they do not disentangle forward and backward causality, ie whether differential expression is a cause of or is caused by the counterfactual expression of some phenotype. It may be that transcriptional response to both asthmatic lung stress and exercise-induced lung stress is similar, driving coincident differential expression. Of the latter, it also does not distinguish between pathological or adaptive responses to disease – whether the DE drives or mitigates disease symptoms. Finally, 3) the TWAS methods our predecessors used did not condition on the set of all other body tissues, and thus may reflect an interdependent multi-tissue response driven by some unknown causal tissue. Here, we are less concerned, given the broadly independent patterns we observe in exercise-induced differential gene expression.

Addressing some of these limitations would require richer data than available here, but others may be accommodated by further analysis. To that end, we now suggest in the discussion that future studies may benefit from our work by evaluating specific loci therein for GxE interactions within large-scale human population biobanks:

“Finally, we expect that future studies may benefit from our work by evaluating specific loci therein for GxE interactions within large-scale human population biobanks. Combined, MoTrPAC’s EET study provides a large-scale, cross-tissue map of changes in exercise adaptation that enables generating new mechanistic hypotheses on the disease impacts of exercise training.”

Additionally, if “differentially expressed genes reflect disease-induced rather than disease-causing changes in the transcriptome”⁷, it may be more appropriate to say that the differential expression effects of long-term exercise are similar to those of specific diseases without manifesting those diseases at rest. Rather, exercise stresses the body in ways not unlike disease stress (taxing the lungs and the heart), and adaptation to long-term exercise follows similar transcriptional pathways to adaptation to disease, ultimately yielding a preventative effect when the adaptive response from exercise is able to curtail disease pathogenesis. We agree that this interpretation was far from clear in the original text, so we have modified all our descriptions of transcriptional directional effects from “protective vs. anti-protective” to “more vs. less disease-like”. We have also added the following sentence to the discussion:

“By subjecting the body to disease-like stresses, regular exercise elicits adaptation to the symptoms of those diseases, reducing the risk of their manifestation from the disease itself.”

Comment: (1b) Tissues: It is an even further extrapolation to test across all tissue types and diseases, many of which likely have no relationship to exercise. Notably, GTEx is from older deceased donors, and known to have issues with expression - see for example McCall 2016 AJHG. The authors use heritability of gene expression which is population / cohort specific.

Response: GTEx is one of the few human studies that support genetic analysis across a broad range of tissues. The study by McCall et al, while interesting, is of limited relevance to our analysis by extensive factor correction of GTEx data for each tissue type removing both known and hidden technical factors. In a similar vein, cell lines are often used as experimental systems despite not reflecting the gene expression states of their primary analogues. However, *Caliskan et al*⁸ noted that despite transformation, genetic regulation is reflective of the primary cell state. Here, we have focused exclusively on genetic regulation of GTEx genes and not observed expression levels, and we expect discovered QTLs to be reflective of the underlying

tissue and cell types. We recognize the limited portability of genetically regulated gene expression models as GTEx is mostly European-descendant individuals, and clarify this limitation in text. Additionally, we make a further note of the sex- and age-structure of GTEx, which skews older (and potentially reflects idiosyncrasies of later reproductive transitions, eg into menopause and may mask earlier developmental eQTLs). Specifically, we add the following text to our Discussion:

“Comparison of exercise-independent age and sex effects, meanwhile, may be limited by differences in age between individuals in either sample, as most humans in the GTEx v8 dataset were aged 50+^{9,10} while trained F344 rats were uniformly under eight months of age and therefore well under the age of onset of the F344 rat equivalent of sex-specific, aging-related changes such as menopause¹¹. These results may also have limited portability to non-European populations, as the GTEx sample comprises mostly European-descendant individuals.”

Otherwise, we are limited by the availability of large, multi-ancestry, multi-tissue, multi-age expression panels, though work to expand the variety of data available is ongoing (eg through dGTEx).

■

Comment: (1c) The conclusion of the paper appears to be that there is widespread enrichment (the point estimates all seem to be > 1) of differentially expressed genes from exercise in rats to PrediXcan genes from human GWAS across all tissues and traits (e.g. Figure 3e). Are the authors sure that the null distribution of overlap is correct? Both PrediXcan and DEG genes could have similar bias. Could a random pairing of tissue / PrediXcan results be used to define a null distribution and then look for relative enrichment?

Response: The patterns in Figure 3a,d-e do indeed reflect pervasive enrichment throughout the DEG-TWAS intersects. Specifically, estimates of our alpha “intercept” parameter were on the order of +0.16 log-odds (posterior mean; $\text{invlogit}(0.16) = 0.54$), with 92% posterior mass falling above 0. Fig. 3a/d/e all incorporated this parameter when constructing marginal posterior distributions:

Biologically, this did not seem unsurprising – given that endurance exercise induces systemic stress throughout the body, imposing metabolic and other demands; that the MoTrPAC study selected tissues in part under the expectation that they would respond to exercise; and that the traits considered drew from a broad range of phenotypic variation, we would find it strange if there weren't identifiable signal shared across tissues and traits / trait categories. That said, these effects were weak – at most shifting a probability of a PrediXcan hit from eg 0.48 to 0.52)

During analysis, we validated the implementation of our model and examined misspecification error in two ways – 1) simulating from the model and assessing posterior output for reasonable calibration properties and data informativeness at empirically motivated sample sizes, and 2)

sampling random genesets without replacement (specifically, by permuting gene indices, so as to preserve patterns of interdependence across tissues). These produced posterior output that tended to not pool across traits or tissues in simulation, eg:

Data of this sample size were quite informative in this regard – posterior variance in α was on the order of 0.01, in contrast to a prior variance of 1 (from our standard normal). In both of the above cases, the model is quite confident that whatever effect exists, it is very small.

However, we did not systematically carry out an extensive simulation study to more precisely quantify misspecification error, nor narratively document all steps of the Bayesian workflow¹² as we believed to be outside the scope of the study, with extensive supplemental materials of limited interest to readers (eg see the 82-page example here¹³) beyond providing well-documented,

reproducible code for all analyses performed and reporting on more common MCMC diagnostics. During model development, we briefly explored through simulation experiment some of the statistical properties of posterior distributions generated from meta-analyzing overlapping aggregated Bernoulli count data, and found that when both marginal rates were common, posterior distributions regressed to a mean of 0:

and that when one of the marginal counts became rarer, the approximation improved:

Another minor point of reassurance might be that, despite our model not explicitly incorporating sparsity (though eg sparsity priors, like spike-and-slab distributions or horseshoe distributions), posterior output nevertheless reflected the circumstance where most tissue \times trait combinations were broadly uncertain in their directionality of enrichment, with a smaller number showing stronger confidence:

The above histogram is not interpretable in quite the same manner as a p-value histogram (where eg p-values are anticipated to be $\text{Uniform}(0,1)$ distributed under the null), but averaged across replicates drawn from the prior, the frequentist properties captured by a similar histogram constructed relative to each parameter's true value should be $\text{Uniform}(0,1)$ distributed, representing the state of perfect calibration (a credible interval of given probability mass covering the true value of the model parameter the appropriate proportion of time).

■

Comment: (1d) MoTrPAC gene expression: the low concordance of differentially expressed genes across tissues - even those of skeletal muscle - is surprising. Is this due to cell type differences, scaling or normalization, or a result of some other issue? For example, it seems like LDLR might change in a similar direction in all tissues even if the magnitude is different. This

result is also confusing in light of the plot in the supplement which appears to show positive correlations between differential expression at different weeks between tissues? Perhaps, this information is in reference 4 (the 'main' manuscript?) which is not available to the reviewer.

Response: In part, the level of sharing is a conservative estimate that is impacted by thresholding. In our study, we focused on the subset of DE genes that were concordantly changing across study (i.e. changed by Week 8). We do expect that there will be some tissue-specificity and sharing to the results and this is indeed a part of the main paper. As noted for reviewer 1, we apologize that the citation was not complete in our original submission and provide a reference¹ and link to the preprint on bioRxiv. It has recently been provisionally accepted.

■

Comment: (1e) The rationale for choosing Week 8 expression should be clarified. Would a model that maximized the data and correlation among samples be superior to identify exercise induced change, consider for example `duplicateCorrelation()` in limma to address the relatedness among samples.

Response: We chose Week 8 to represent the exercise-adapted state as it was the last week of the MoTrPAC PASS-1B training protocol, and so represented the longest-term transcriptomic changes from exercise training, as opposed to more ephemeral effects (eg unadapted inflammation response) at earlier collection weeks. We have clarified this decision in text, amending the line:

“All samples were harvested 48 hours after the last exercise bout, and the 8-week time point was taken to correspond to the adapted state”

to instead read:

“All samples were harvested 48 hours after the last exercise bout, and the 8-week time point was taken to correspond to the adapted state, as it allowed for the greatest degree of long-term adaptation to exercise to have occurred, as well as the least degree of unadapted acute response (eg inflammation).”

We chose not to incorporate results from earlier timepoints because we had wanted to focus exclusively on these long-term adaptation effects.

■

Comment: (2a) Please further describe the calculation used for “these empirical priors”. The current description provides only an overview of the approach - the calculation may involve the sample variance of expression level, or a component from the method of moments?

Response: The exact procedure used is described below. We initially tried several more cohesive modeling strategies, but ultimately did the following for reasons of computational tractability:

1. Modify the gtex-pipeline using the GTEx v8 data release for each gene in each relevant tissue to use a $\log_2(x+1)$ transformation instead of an inverse normal transformation. This was taken as an approximation to averaging over uncertainty in some unobserved mean rate (eg, as from a gamma-poisson or lognormal-poisson, offset with library size) for each individual, which was found to be computationally intractable.
2. Fit a linear model (via OLS) to these values independently for each gene and tissue using GTEx v8 covariates. “Regress” or “residualize” out the estimated effects of these covariates by subtracting from each individual the values corresponding to their covariates.
3. For each gene in each tissue, compute a sample variance (implicitly within-sex, as sex was a covariate in the above step).
4. For each tissue, fit an inverse gamma distribution across gene variances. Using these fitted values, update this “empirical” prior using each sample variance and sample size, separately for each sex. Fitting the inverse-gamma separately for each sex would have yielded fully sex-specific estimates, but we estimated pooled hyperparameters and unpooled posteriors to approximate regularization of each sex-specific variance towards a pooled mean variance.
5. Compute the expectation of these sex-, tissue, and gene-specific posterior distributions as an estimate of total phenotypic variance stratified along the above axes.

We apologize for not including these details or motivations in the written methods, but had endeavored include many of these methodological comments and documentation in our provided code for reproducibility. We have also included the above text in a Supplemental Methods section, adding the following line to the relevant location of the Methods section of our main text:

Additional details are provided in the Supplemental Methods.

■

Comment: (2b) The standardization process described at line 439-447 is confusing - is there information on the theory or reference? It is difficult to understand why fold changes are divided by the estimated standard deviation of gene expression level. This is an important concept in this article and needs to be specified.

Note for this section, Figure 1c and Figure 1d appear to be incorrectly referenced.

Response: Apologies for the mistaken referencing – it has been amended. The standardization we applied did not leverage any quantitative genetic theory beyond the definition of SNP-heritability, $h_{2\text{SNP}} = V_{\text{SNP}}/V_P$. Specifically, we wanted to understand whether a given amount of differential expression corresponded to a small or large amount of natural variation – a change of 2^3 in some gene's expression may be massive if individuals hardly vary in expression, or trivial if inter-individual variation spans multiple orders of magnitude. Given both genetic (from inbreeding) and environmental (from standardized housing, enrichment, etc. not corresponding to the exercise intervention) homogeneity among F344 rats, traditional standardized metrics such as Cohen's d or Hedges' g would not be interpretable on the scale of interest, so we instead standardized according to tissue-specific variation attributable to both SNP and total effects in the GTEx population.

Thank you for catching this error – we have corrected those figure references in text.

■

Comment: (2c) Please clarify the dimensions for defined vectors or matrices in Table S2 (With predefined number of genes and hits for i and j).

Response: We have added a column to Tables S2 and S3 specifying the dimension of each data or model variable.

■

Comment: (3a) Clarity in text: The analytic pathways and assumptions are complex. It is not clear from the text, for example, whether the author is discussing rat or human datasets. The figure does make things clearer, but the text should match. An effort should be made to increase readability, e.g. one of the figure captions reads, "... denominator lie a stacked his-

togram of h2 SNP estimates for expression scores from GTEx with a p-value > 0.05 after IHW correction, and the collection of inverse-gamma distributions serving to regularize log2 normalized expression scores after covariates had been residualized out.”

Response: Thank you for the feedback. To clarify whose genes are being discussed, we have added the following sentence at the start of our Results: “Exercise training induces differential expression of rat genes across multiple body tissues, and many of these genes can be mapped to human orthologs: 94.5% of all unique DE genes (87-98% across tissues), and 79% of all expressed genes (85-93% across tissues)”, and also modified the following sentence to read “...in the subset of rat genes with identifiable human orthologs (hereafter ‘genes’, unless otherwise noted)”.

To further improve readability throughout the text and impartially identify low-readability sentences, we have applied the Flesch Reading Ease Index to each sentence of the manuscript. Specifically, we extracted the manuscript to plaintext via *OpenDetex*¹⁴ and processing tokenized sentences via the `textstat_readability()` function in the *quanteda.textstats* package¹⁵ in R, flagging all sentences scoring less than 10 for manual revision. 51 / 167 sentences were identified by this procedure. While many of these were “false alarms” (eg lists of tissues, grant #s, etc.), we agree that several of the remainder, including the one identified above (score = 3.1), would benefit from revision. We have amended them as follows:

Original:

Despite human-rat differences, our unbiased approach identified multiple results recapitulating established exercise biology and disease relationships; however, several were surprising.

Amended:

Despite human-rat differences, our unbiased approach identified multiple results that echo established exercise-disease relationships. However, some findings were unexpected.

...

Original:

Future causal inferential work may use the genetic correlates of physical activity as instruments to infer tissue-specific drivers of phenotypic adaptation, but analysis of experimental data from animal models will remain an important complement towards informing our understanding of how exercise impacts gene expression in different tissue and organ systems and influences a large variety of human traits and diseases.

Amended:

Future causal inferential work may use the genetic correlates of physical activity as instruments to infer tissue-specific drivers of phenotypic adaptation in humans. But analysis of experimental data from animal models will complement these efforts where genetic effects are weak (Fig. 1d), targeting causality directly to identify how tissue and organ systems adapt to exercise and influence a large variety of human traits and diseases.

...

Original:

But while these biological differences limit direct interpretation of orthologous exercise-induced gene regulation, research at the degree of tissue resolution and experimental compliance enabled by a rat model is essential to furthering our understanding of the molecular transducers of physical activity in tissues that are not easy to biopsy in subjects who are less easy to motivate.

Amended:

But while species differences can complicate interpretation of exercise-induced regulation of orthologous genes, these models remain crucial and provide high-levels of experimental compliance and tissue accessibility.

...

Original:

As the landscape manuscript observed across all rat genes, we found that after long-

term exercise training, there was little interdependence in gross transcriptomic adaptive response across tissues in the subset of rat genes with identifiable human orthologs (hereafter 'genes', unless otherwise noted), with only two pairs of tissues in females - the skeletal muscles vastus lateralis and gastrocnemius, as well as white adipose and the colon - producing Spearman's ρ 's at a level greater than 0.3 (Fig. b).

Amended:

As observed in the main MoTrPAC PASS1B paper, we found that after long-term exercise training, there was limited overall concordance of adaptive differential expression across tissues in the subset of rat genes with identifiable human orthologs (hereafter 'genes', unless otherwise noted). Only two pairs of tissues in females - the skeletal muscles vastus lateralis and gastrocnemius, as well as white adipose and the colon - produced Spearman's ρ at a level greater than 0.3 (Fig. S1b).

...

Original:

Here, we visualize conditional heritability enrichments in differentially expressed, sex-homogeneous genesets corresponding to different tissues across the set of considered traits, as estimated by LDSC.

Amended:

Here, we visualize conditional heritability enrichments (LDSC) of multiple traits within differentially expressed, sex-independent genesets corresponding to different tissues.

...

Original:

While exercise-responsive genesets were enriched for PrediXcan genes associated with cardiometabolic traits (Fig. 3e,f), and members of this intersection appear to broadly point in the protective direction (Fig. 4), we also found a risk enhancing effect among genes in the intersection for asthma and body fat percentage (Fig.

4), though these did not appear to have intersection sizes greater than expected by chance, with the latter showing only weak evidence for depletion (Fig. 3f).

Amended:

Genesets that responded to exercise were enriched for PrediXcan genes linked with cardiometabolic traits (Fig. 3e,f). The intersection of these genes seems to lean towards protective effects (Fig. 4), but we also found risk-enhancing effects for genes associated with asthma and body fat percentage (Fig. 4). These associations, however, did not exhibit intersection sizes larger than expected by chance, and the latter showed only weak evidence of depletion (Fig. 3f).

...

Original:

Accelerated rat life history also serves to make experiments on adaptation to exercise training more feasible given the timescales relevant to the subject lifespan, and rat behavior is easier to regulate than human behavior, eliminating biases related to non-adherence and attrition.

Amended:

Accelerated rat life history makes it more feasible to conduct experiments on exercise training adaptation on timescales relevant to their lifespan. Moreover, it's simpler to regulate rat behavior than human behavior, reducing biases linked to non-compliance and attrition.

...

Original:

On the horizontal axis, we represent estimates conditional on an annotation that includes all other tissue-specific annotations alongside the baseline annotation of 53 functional categories.

Amended:

The horizontal axis represents estimates that are conditional on an annotation that includes all other tissue-specific annotations and a baseline annotation of 53 functional categories.

...

Original:

Taking the posterior mean of inferred expression variances, we computed their square root to obtain an estimated standard deviation, which we used to divide estimated exercise DE to produce standardized estimates in units of within-tissue phenotypic standard deviation (SD), conditional on sex and population (quantile plots in upper panels of Fig. d).

Amended:

For each gene, we then took the square root of the posterior mean of inferred expression variance to estimate within-population standard deviation of expression. This value was then divided by the estimated exercise DE to produce standardized estimates in units of within-tissue phenotypic standard deviation ($SD_{phenotype}$). Further, these estimates were conditioned on both sex and population (quantile plots in upper panels of Fig. 5d).

...

Original:

As these estimated genetic correlations were done on a pairwise basis, positive-semidefiniteness is not guaranteed, so we instead substituted the nearest PSD correlation matrix using Higham's algorithm, implemented in the R-package Matrix function `nearPD()`.

Amended:

As these genetic correlations were estimated pairwise, positive-semidefiniteness (PSD) of the whole correlation matrix is not guaranteed. To satisfy the PSD constraint of G_{SNP} , we substituted the nearest PSD correlation matrix output from

Higham's algorithm⁶, implemented in the R-package *Matrix*⁷ function `nearPD()`.

...

Original:

At the level of individual traits, we observed several confident enrichments across a range of cardiometabolic markers (primarily cholesterol and saturated fatty acids), as well as thyroid disorders and multiple sclerosis.

Amended:

We also noted several specific trait enrichments across various cardiometabolic markers, mainly cholesterol and saturated fatty acids, as well as thyroid disorders and multiple sclerosis.

...

Original:

Examining directionality of trajectories for 8w genesets corresponding to the two non-anthropometric traits in these results, we observed a regression towards a mean proportion of 0.5 across tissues, likely the product of underlying genes no longer being differentially expressed at earlier timepoints (Fig. 5).

Amended:

When examining the direction of trajectories for 8-week gene sets linked to the two non-anthropometric traits, we noticed a regression towards a mean proportion of 0.5 across tissues. This is likely due to underlying genes only being differentially expressed at later time points (Fig. 5).

...

Original:

Pseudo-replication across tissues and traits amplifies signal visible to higher-level

parameters, causing inference of the latter to mistake interdependent effects as independent evidence for enrichment.

Amended:

Pseudo-replication across tissues and traits amplifies signal visible to higher-level parameters, leading the inference model to mistake interdependent effects as independent evidence for enrichment.

...

Original:

As aggregating to a Binomial removes all signal of gene interdependence, we introduce parameters Σ_i and Σ_j , corresponding to $i \times i$ tissue and $j \times j$ trait correlation matrices.

Amended:

When aggregating many Bernoulli random variables to a single binomial, signals of gene interdependence that would otherwise prevent this are lost. To address this, we introduce parameters Σ_i and Σ_j , corresponding to $i \times i$ tissue and $j \times j$ trait correlation matrices, respectively.

...

Original:

Further, there was very little overlap in differentially expressed (DE) genesets (DEGs) corresponding to each tissue (Fig. 1b)- approximately 78% of DE genes were unique, differentially expressed in only one tissue, with 95% of genes matching at most a pair of tissues, and only one pair of tissues demonstrating a Jaccard index > 0.1 (the gastrocnemius and the vastus lateralis, Jaccard Similarity ≈ 0.21).

Amended:

Further, there was little overlap in differentially expressed (DE) genesets (DEGs)

corresponding to each tissue (Fig. 1b). Approximately 78% of DE genes were unique and differentially expressed in only one tissue, and 95% of genes matched at most a pair of tissues. Only one pair of tissues showed a Jaccard index > 0.1 (the *gastrocnemius* and the *vastus lateralis*, Jaccard Similarity ≈ 0.21).

...

Original:

Adopting a previous approach to identify environmental modulators of genetic risk factors, we aimed to assess if exercise modulated any specific traits or diseases.

Amended:

We investigated whether exercise specifically modulates any traits or diseases by building on a previous approach to identify these effects.

...

Original:

In its denominator lie a stacked histogram of estimates for expression scores from GTEx with a p-value 0.05 after IHW correction, and the collection of inverse-gamma distributions serving to regularize log normalized expression scores after covariates had been residualized out.

Amended:

A stacked histogram of estimates for h_{2SNP} for expression scores from GTEx (p-value < 0.05 after IHW correction) is on the left in the denominator. On the right are the inverse-gamma distributions serving to regularize log₂-normalized expression scores.

...

Original:

The horizontal axis, corresponding to a given quantile in (0,1), was logit-transformed; the vertical axis, corresponding to the ratio of DE / SD(log₂expression), received an inverse hyperbolic sine transformation.

Amended:

The horizontal axis, which corresponds to a given quantile in (0,1), was logit-transformed. The vertical axis, corresponding to the ratio of DE/SD(log₂ expression), received an inverse hyperbolic sine transformation.

...

Original:

As signal for the directionality of effect (protective or risk-enhancing) cannot be shared across traits in the same manner as their overall frequency, there is limited ability to pool the location of trait difference effects across traits within a given category, or across all trait altogether.

Amended:

Unlike for their overall frequency, signal for the directionality of effect (more or less disease-like) cannot be shared across traits or within trait categories, as the traits themselves vary in whether they are harmful, neutral, or beneficial.

...

Original:

When excluding easily biopsied tissues such as blood, the skeletal muscles, and adipose, the number of well-established disease genes associated with adaptive exercise training response was 178 across 143 traits, including 101 traits with no gene-trait associations in any easily biopsied tissue.

Amended:

When we excluded easily biopsied tissues such as blood, skeletal muscles, and adi-

pose, we found 178 well-established disease genes associated with an adaptive exercise training response. This corresponded to 143 traits and included 101 traits without any gene-trait associations in an easily biopsied tissue.

■

Comment: (3b) Clarity in figures: Too many similar colors that don't have meaning make it hard to interpret. For example, what is the message of Figure 1b (lower half)? Would consider simplifying: instead of trying to present all tissue types / GWAS / etc. consider limiting and / or grouping and using Tables to show all data.

The number of trait x gene pairs will be affected by the number of DEGs. This could potentially make it difficult to compare values across the lines.

Response: We apologize for any difficulties distinguishing colors. Early in the process, we explored methods to maximize separability of discrete color choices, but ultimately opted to use the same colors as the main manuscript for greater consistency. Fig 1B specifically attempted to graphically summarize the distribution of DEGs across tissues, while also presenting numeric counts. Stacked histograms are even less legible when counts in each bin vary substantially, so we included both the gray histogram to show the number of DEGs shared by two, three, four, etc. tissues, and the proportional stacked histogram to show the composition of those pairs, triplets, etc. The bottom panel shows that overlap in gene sets across tissues, even if weak overall, is not driven exclusively by a small subset of tissues (eg the two skeletal muscles).

We also agree on the second point. It's also affected by things like GWAS sample size (influencing power to detect trait x gene associations) – we aimed to design our later analyses in a manner that accommodated these effects.

■

Comment: (4a) The apparently random selection of GWASs seems to result in increased noise without clear signal. There are results such as - 4 DEGs in the hippocampus related to body fat percentage and 9 colon genes related to asthma? It seems that expanding the list of metabolic traits would make more sense.

Response: Our selection of GWAS was meant to serve three purposes – 1) to tie into previously uniformly processed published work in the transcription-phenotype literature, 2) to serve as

both “positive” and “negative” controls, recapitulating known exercise biology and helping to locate it to inaccessible candidate tissues over unexpected or unlikely results, while 3) allowing opportunity to identify novel associations across a broad cross-section of human phenotypic variation. We do not intend for the associations identified here to have any final say in identifying the mechanisms by which exercise structures human trait biology, but rather to serve as a springboard for both future exploratory work and to potentially corroborate surprising results independently obtained elsewhere.

■

Comment: (4b) The discussion uses weak data and speculation. It is unclear how to relate human height effects which are likely relevant to childhood / growth, in adult rats that are exercising. Worse, to suggest that exercise causes gene expression that mimics and therefore worsens asthma based on splenic (and kidney and hippocampus - Figure 5) expression, and genes with weak evidence of genetic causality (references 54-57), is not credible and is a potentially dangerous public health message. Please modify.

Response: We have reviewed and attempted to tone down causal language in the discussion, changing the following:

“Asthma also emerged as a trait that adversely affects exercise.”

to:

“Asthma also emerged as a trait with shared transcriptional effects as exercise.”

However, the results on asthma should not be viewed as a “dangerous public health message”. There is in fact ample literature on exercise-induced asthma and exercise-induced airway dysfunction already - this is a known medical condition.

For human height effects, we agree this is more speculative as height is highly heritable (heritability estimated at $\approx 80\%$), and it has long been a myth that exercise in kids impacts their statural growth. As such we have made this clearer by making this change. As such, we have edited:

“While evidence for exercise effects on height is equivocal”

to now read:

“Evidence for exercise’s effect on height is weak and ambiguous. However, ...

We have also made extensive additional edits to temper the causal language and interpretation used throughout this work (see below).

■

Comment: (4c) The directionality is confusing: There are “positive” effects for BMI, asthma, and “negative” effects on height? This seems to suggest that exercise would increase BMI and body fat and asthma; and reduce height? Is this correct?

Response: Our position is that animal models are not able to fully supplant clinical study, while also enabling research that would be very difficult and less ethical to perform in humans. While rat and human biology is not fully commensurable, we believe it is close enough to provide motivation and opportunity for later work. We also apologize for the confusion, which potentially resulted from a desire for greater brevity that stripped some of the original text of more nuanced causal interpretation. Specifically, our understanding is that while S-PrediXcan and related methods do, in principle, identify causal relationships between traits and expression, they do not separate the direction of expression \leftrightarrow trait causality or accommodate correlated expression, while bidirectional and multi-tissue methods are lower powered. Thus, at this level of resolution, we are only able to make claims of shared biology between the exercise intervention and complex traits. Careful experimental interventions such as MoTrPAC’s long-term Preclinical Animal Study block all non-random causal input into exercise, so we know that differential gene expression lies downstream of exercise behavior. With traits such as asthma that we devoted most of Discussion to discussing, it may be that similar patterns of gene expression like upstream or downstream, causing or being caused by the phenotype of interest.

Another interpretation is that exercise and asthma both elicit similar transcriptional responses, perhaps in response to increased respiratory difficulty. If the overlap were more extensive, asthma could be compared to having the respiratory difficulties that result from cardiorespiratory exercise without actually having exercised. If someone approaches you looking like they just ran a marathon while having only walked across the room, you might suspect the presence of some disease process, without marathon-running being upstream of that disease. Exercise is acutely damaging, taxing, and fatiguing, but it is by adapting to that mindfully titrated dam-

age under conditions of appropriate nutrition and opportunity for recovery that we benefit, and specifically achieve health benefits while at rest. In contrast, diseases are often unrelenting and tend not to provide well-spaced rest periods.

Finally, BMI is also influenced by muscle mass and not just fat mass. These components are challenging to dissect here.

Space constraints resulted in the omission of text such as the above in our Discussion, but the above comments have motivated its re-inclusion. Specifically, we add the following:

“By subjecting the body to disease-like stresses, regular exercise elicits adaptation to the symptoms of those diseases, reducing the risk of their manifestation from the disease itself.”

To further clarify this more nuanced causal interpretation, we have modified all our descriptions of transcriptional directional effects from “protective vs. anti-protective” to “more disease-like vs. less disease-like”.

■

Comment: (4d) There are several confusing aspects of the chosen traits (why are there two asthma GWASs, and why is one Allergy the other as Autoimmune)? Why was pulmonary embolism, and father’s age at death chosen?

Response: We relied on previously published work⁴ for the set of traits that we ultimately used, preserving those authors’ original annotations except in a single case where there seemed a clear coding error (“Multiple_Sclerosis” was miscoded as “Cardiometabolic”, with the citation provided appearing unrelated to that topic, so we re-coded it to “Psychiatric-neurologic”). We did this to preserve comparison with this and other literature (eg Gay et al 2020¹⁸). Fundamentally, we wanted to capture the breadth of human phenotypic variation, using similar uniformly processed GWAS as a check for obtaining similar downstream results, and GWAS unrelated to exercise as negative controls.

■

Comment: (M1) Figure 1d: “h² SNP estimates for expression scores from GTEx with a p-value > 0.05” so all non-significant heritabilities were included? The plot also seems to show that estimated h² SNP - most are between 80-100%?

Response: Our apologies – that was a typo, and should read “p-value < 0.05”. LDSC authors recommended and prior work¹⁹ used GCTA to estimate tissue-specific SNP-heritabilities. As the latter used inverse-normal transformed expression and allowed unconstrained heritabilities (allowing negative – and therefore impossible, as variances are strictly positive – values), we performed our own analysis, but most of the output had point-estimated SNP-heritabilities at or near 0. This resulted in genetic (SNP) variances also near 0, and therefore standardized DE effect sizes at $\pm\infty$, which – while potentially correct – obscured the relationships we were trying to show (the distribution of genes whose genetic effects on expression exercise could overcome). So we significance-filtered these tissue-x-gene heritabilities down to those “significantly” nonzero as output by GCTA for plotting, which required larger effect sizes at similar variance to pass the filter.

To make this clearer, we have altered the following text in our Methods, changing:

“Several of these h_{2SNP} point estimates were at or near 0, resulting in extreme standardized effect sizes. As a further filter, we thresholded on significance (IHW $\alpha = 0.10$, with tissue as a covariate) to focus on confidently heritable genetically regulated expression.”

to:

“Many of these h_{2SNP} point estimates were at or near 0, resulting in extreme standardized effect sizes. As a further filter, we thresholded on significance (IHW $\alpha = 0.10$, with tissue as a covariate) to focus on confidently heritable genetically regulated expression. This removed $\approx 92\%$ of gene x tissue pairs (583,238 / 632,738), leaving 49,500 for later analysis and use in figures.”

■

Comment: (M2) Consider using exercise induced bronchoconstriction instead of exercise induced asthma.

Response: We have amended the following lines in light of this suggestion:

Original:

“This may be due to shared etiology between the general asthmatic condition mea-

sured by self-report and exercise-induced asthma (EIA)... triggers an inflammatory response alongside bronchoconstriction... cell types play in structuring EIA.”

Amended:

“This may be due to similar etiology between the general asthmatic condition measured by self-report and exercise-induced bronchoconstriction (EIB)... triggers an inflammatory response alongside shortness of breath... cell types play in structuring EIB.”

■

Comment: (M3) Intracorneal volume (typo).

Response: Thank you for catching this error. We have corrected it in all figures in which it appeared.

Reviewer 3

Comment: (M1) Line 779. Please provide your meaning of the word “compliance” in line 779.

Response: We have clarified the meaning of this term, amending the line to read: “But while species differences can complicate interpretation of exercise-induced regulation of orthologous genes, these models remain crucial and provide high-levels of experimental compliance and tissue accessibility in subjects who are far more straightforward to motivate.” The idea being that the animal protocol allowed personnel to eg “use a light shock to make [the rats] run [only for those rats that will not run continuously]”, an incentivization mechanism not used in human participants for ethical reasons.

■

Comment: (M2) Justify with references your usage of a single time point at 48 hours post exercise? You should provide other time points. For example, PGC1alpha transcription peaks at zero hours post exercise and PGC1Alpha mRNA peaks at one-hour post-exercise, is still increased 6 hrs post exercise and unchanged 24 hours post exercise in a biopsy from the middle portion of the vastus lateralis muscle of each leg from 19–24-year-old, male subjects in a paper cited over 664 times (Pilegaard et al. *J Physiol.* 2003 Feb 1;546(Pt 3):851-8. doi: 10.1113/jphysiol.2002.034850.). (See general comments #4 in the next section)

Response: A 48 hr time point was chosen for the main study as it was also the last time point to be used in the acute preclinical phase of MoTrPAC, which will examine the short-term effects of exercise at much finer temporal resolution than this phase of MoTrPAC studies. Additionally, external studies show gene expression returning to baseline 48h after exercises (Sue Bodine, personal correspondence).

■

Comment: (M3) Inconsistency from Abstract to main text. The authors should carefully examine and proofread how many tissues were included in the study and their descriptions are consistent across Abstract, figures, and main text (e.g., it was mentioned 15 tissues were analyzed in Abstract but described 19 tissues were used for analysis in Methods)

Response: We apologize for the inconsistency – we have clarified this in text (the discrepancy arose from the main manuscript, which includes eg blood plasma as a tissue, and from excluding each of the separate gonads).

■

Comment: (G1) The main purpose of the study appears to be to discover the impact of exercise on trait genetics. While integrating human disease traits and the alterations of gene expressions in rat exercise training, it is highly possible that some of the human disease-related genes may not be translatable from rat orthologs to human gene sequences. We suggest that the authors should expand a single sentence in the discussion for the potential lack of obsolescence in translating rat sequences to human sequences at the sites of transcription factor bindings.

Response: We have added sentences to describe the orthology mapping.

In the Methods, we added the following sentence:

“To enable comparison between genes expressed in rats and humans, we compiled a MoTrPAC rat-to-human ortholog map from GENCODE and RGD resources^{1,20,21}.”

In the Results, we add the following clause following the first comma:

“Exercise training induces differential expression of genes across multiple body tissues, and many of these genes can be mapped to human orthologs: 94.5% of all unique DE genes (87-98% across tissues), and 79% of all expressed genes (85-93% across tissues).”

In the Discussion, we add the following sentence:

“Identification of rat-human gene orthology is also a difficult problem, and important biology almost certainly lies within disease and exercise-responsive genes across species whose correspondence could not be established.”

If it's any reassurance, prior to submission we explored sensitivity to orthology identification by re-running our workflow using a PANTHER-derived gene map and found very little quantitative and no qualitative difference in results.

■

Comment: (G2) It has been well-established that male/female differences have an essential role in driving exercise adaptation (J Physiol. 2023 Feb; 601(3):419-434. Epigenetics 2019 Jun; 14(6):523-535). The candidate genes used in the study were the genes that went up in both

sexes. As some genes respond in opposite directions in males and females to endurance-type exercise, the criteria used in the study may lose sex-specific gene regulation after exercise training. If no further sex-specific analysis are to be done, please address sex differences in the discussion as a limitation of the current study.

Response: Thank you for the suggestion and references. We agree that sex (as well as age, nutrition, etc.) can have profound effects on exercise adaptation, which is made more difficult by the GTEx sample (most female humans in the GTEx v8 dataset were aged 50+^{9,10} and thus potentially menopausal, while trained female F344 rats were uniformly under 8 months of age and therefore well under the age of onset of menopause¹¹). We'd initially investigated sex-specific DE across rats and humans and found limited replication, so we tried to strike a balance between examining sex-independent effects by limiting ourselves to genes that were DE in both sexes, while also acknowledging differential magnitude in that DE by incorporating sex as a covariate when assessing standardized effect sizes. In light of your suggestion, we have added the following sentence to our discussion:

“Qualitative sex-specificity, a notable hallmark of exercise adaptation in humans^{22,23}, also fell outside the scope considered here, though is afforded closer treatment in companion publications²⁴.”

Comment: (G3) The protective/anti-protective effects of exercise training can depend on whether exercise is aerobic or is resistance type of exercise. We suggest that the authors should identify the term exercise in the Abstract and Discussion by using either “endurance exercise” or “aerobic exercise” to help readers to specify the findings in the current study are from endurance exercise training.

Response: Following this suggestion, we have modified the first sentence of the Abstract to read “Endurance exercise training is known to reduce risk for a range of complex diseases.” and the Introduction to read “Endurance exercise is associated with multiple positive health outcomes.” to maximize consistency with other MoTrPAC publications².

Comment: (G4) While using the term “exercise response”, it should be noted that under the current study design for sample collections (48-hour after last training period), the results would

be more related to “baseline gene expression after exercise training” or “gene expression in resting tissues after exercise training”, which in other words, it is a response to exercise adaptation, but not a response immediately after exercise. “Exercise adaptation” and “exercise response” may not be interchangeable since one refers to the training effect and the other refers to response after a bout of exercise. This may explain the results why fewer alterations were found at 1-week and 2-week time-point, since the effects of exercise were recovered from the last bout of exercise and exercise adaptation had not yet fully developed. Some mRNAs only change during the exercise-bout itself. Others only change for a few hours after exercise. For example, human PGF1alpha does not change during exercise, but PGC-1alpha is increased 2 hrs post-exercise, and then is at resting levels at 10 hrs post exercise. This point mirrors major concern #2.

Response: We agree with this clarification and interpretation of MoTrPAC EET results. To further complicate matters, there is likely also a long-term adaptive component to acute response (eg there is a training effect on acute responsivity to endurance exercise in skeletal muscle AMPK activation²⁵). To clarify in-text, we have made the following insertions:

“after long-term exercise training, there was limited overall concordance of adaptive differential expression across tissues in the subset of rat genes with identifiable human orthologs”

“number of well-established disease genes associated with adaptive exercise training response”

“PrediXcan-significant genes overlap adaptive training-response genes.”

“we modify the hive plot concept to depict a Spearman correlation network of adaptive transcriptional response to long-term exercise across timepoints”

■

Comment: (G5) Data visualization: Since the scope of the study is to investigate the influence of exercise training in gene regulation with inter- and intra-tissue levels, we recommend either using colorblind-safe labeling or adjusting the choice of colors in the current figures. Some of the color themes used in the current figures may be difficult to distinguish and interfere the interpretation of results from readers. For example,

Fig. 2: BLOOD, HIPPOC, and LIVER

Fig. 3a and Fig. 4: color theme for the X-axis

Fig. 4: HIPPOC, HYPOTH, SPLEEN, and HEART

Response: We wholeheartedly agree with this criticism, and typically strive to both apply colorblindness simulators (eg colorblindr) to our own data visualization, as well as incorporate colorblindness as a parameter when developing our own tools used to discretize colorspace. At the dimensions seen here (15 tissues + 12 trait categories), however, we've found it difficult to generate palettes with nicely separable colors. For the axis in Figs 3a and 4, we used the *Dark2* palette and (supplemented by the two most distinct entries in *Set2*) from *RColorBrewer*²⁶, which is listed as "color blind friendly" (ie, it is visible at `RColorBrewer::display.brewer.all()` with `(colorblindFriendly=TRUE)` toggled). For the tissues, we used the color palette used throughout all MoTrPAC publications, which our own publication is not able to alter and must adhere to for consistency.

In light of your suggestion, however, we have tried to incorporate double encoding to trait category color legends in Figures 3 and 4, including category abbreviations – Anthropometric (An) Blood (Bl) Psychiatric (Ps), etc. – on all relevant figures.

■

Comment: (MR1) The source of the data and the methods from MoTrPAC EET study, which refers to Ref#4 cited in this paper, has not been published while reviewing this manuscript. We suggest that the authors should cite the most recent version of the manuscript on BioRxiv (<https://www.biorxiv.org/content/10.1101/2022.09.21.508770v3>) and the publicly accessible NIH / GitHub from MoTrPAC (<https://commonfund.nih.gov/moleculartransducers>) (<https://motrpac-data.org/data-access>) to help readers to understand MoTrPAC study design and track MoTrPAC related publications extensively.

Response: We apologize for incorrectly citing the main manuscript, which has been provisionally accepted to Nature during this review process. We have updated the citation, also pointing to the public data release. Our aim is that this manuscript will be timed as a companion manuscript to that study and released simultaneously as part of an NPG collection.

■

Comment: (MR2) Line 277-289. The authors briefly described the study design of MoTrPAC EET and the sample collections were from female and male F344 rats in a total number of 50

rats across 19 tissues. However, only 874 samples were extracted. Does this suggest that only 46 rats were in the study ($874 \text{ samples} \div 19 \text{ tissues} = 46 \text{ rats?}$) or there were some missing data from some rats or tissues? As Line 311-316 mentioned, rats under sedentary condition were included as no exercise control in this study and analyses, please address it in the study design section and clarify the number of animals in each condition, time-point, and sex.

Response: We apologize for the confusion. The discrepancy between the two numbers from a few different sources – the main manuscript distinguished between gonadal tissues (excluded here), 16 outliers were detected and excluded from the data before DEA, and tissue contamination between brown adipose and vena cava resulted in their exclusion. Further details can be found in the landscape manuscript. We have amended the corresponding sentence to read “In this work, we leverage data from a total of 738 extracted samples across 15 tissues and 47-50 rats per tissue that were subjected to RNA-sequencing and differential expression analysis”.

▪

Comment: R1. The main findings for Fig. 1D and 1E did not appear in the Results for Fig 1. (Line 87-167) and may confuse readers. Since the identifiers of Fig. 1D and Fig. 1E showed in the later part of the main text (Line 447 and 649) for the first time. The authors need to correct this.

Response: We apologize for the oversight and agree that Figure 1 has too many panels and sub-panels. For thematic clarity, we have split Figure 1 into two separate figures, renumbering all subsequent figures accordingly.

▪

Comment: (R2) Lines 145-167, Page 3. We encourage the authors to use a table to present the results described from Line 145-167 to help readers having a better understanding.

Response: We apologize for the omission, which was caused by our failure to navigate the submissions portal. In case we are again unsuccessful, please find these and other tables in the following directory of the associated GitHub repo: github.com/NikVetr/MoTrPAC_Complex_Traits/tree/main/supplemental_files

▪

Comment: (R3) The overall resolution for Fig 3 is visibly lower than other figures in the manuscript. Please make sure that the description in Line 198-199 is consistent with the heading of Figure

3a.

Response: We apologize for the lower resolution – all figures in the manuscript were vector graphics, and so in principle have “infinite” resolution, but different software and hardware can occasionally fail to appropriately render particular images. We suspect the issue here might be the graduated arrows in Fig 3b,c, whose internal spaces are represented by many tiny polygons. We encountered a similar issue drawing the arrows in 1a, and ended up raster-izing their contents at low resolution before overlaying the vector graphics. We’ve now done the same here. Please let us know if that helped!

Closing Remarks

Finally, we have made a few minor unsolicited revisions. Most of these are stylistic, and are highlighted in the accompanying comparison document. Three more substantive revisions beyond these are: 1) We reworked parts of the associated GitHub repo and have hosted larger files not able to be stored on GitHub on Zenodo. 2) a collaborator noticed that a particular phenotype (*Multiple_Sclerosis*) had been miscoded in previous work (*Cardiometabolic*), with the citation provided²⁷ appearing unrelated to that topic, so we re-coded it (to *Psychiatric-neurologic*). And 3), we made a minor modification to the structure of our first multilevel model, changing the trait \times tissue error covariance term away from having correlation matrix $\Sigma_i \times \Sigma_j$ (or the equivalent matrix-normal specification) to estimating pairwise correlations for all tissue $_i \times$ trait $_j$ pairs. These did not meaningfully impact the fitted model, though together modifications 2-3) did result in HEART entering and *Multiple_Sclerosis* exiting the “confidently” enriched subsets of Fig. 4 (formerly Fig. 3), a change we propagated to the text.

We hope that the responses and modifications described above adequately address your questions, suggestions, and concerns. We believe our manuscript to be much improved in response to your feedback and thank you again for your efforts and insight.

We look forward to further feedback and are happy to provide additional clarification or make further changes if necessary.

Sincerely,

Stephen Montgomery and Nik Vetr,
on behalf of all authors,
School of Medicine,
Stanford University

References

1. Group, M. S. *et al.* *Temporal dynamics of the multi-omic response to endurance exercise training across tissues* en. Pages: 2022.09.21.508770 Section: New Results. Oct. 2022. <https://www.biorxiv.org/content/10.1101/2022.09.21.508770v2> (2023) (cit. on pp. 2, 26, 45).
2. Sanford, J. A. *et al.* Molecular Transducers of Physical Activity Consortium (MoTrPAC): Mapping the Dynamic Responses to Exercise. English. *Cell* **181**. Publisher: Elsevier, 1464–1474. ISSN: 0092-8674, 1097-4172. [https://www.cell.com/cell/abstract/S0092-8674\(20\)30691-7](https://www.cell.com/cell/abstract/S0092-8674(20)30691-7) (2022) (June 2020) (cit. on pp. 2, 46).
3. Ochoa, D. *et al.* Open Targets Platform: supporting systematic drug–target identification and prioritisation. *Nucleic Acids Research* **49**, D1302–D1310. ISSN: 0305-1048. <https://doi.org/10.1093/nar/gkaa1027> (2022) (Jan. 2021) (cit. on p. 3).
4. Barbeira, A. N. *et al.* Exploiting the GTEx resources to decipher the mechanisms at GWAS loci. *Genome Biology* **22**, 49. ISSN: 1474-760X. <https://doi.org/10.1186/s13059-020-02252-4> (2022) (Jan. 2021) (cit. on pp. 6, 41).
5. Wilson, D. J. The harmonic mean p-value for combining dependent tests. *Proceedings of the National Academy of Sciences* **116**. Publisher: Proceedings of the National Academy of Sciences, 1195–1200. <https://www.pnas.org/doi/10.1073/pnas.1814092116> (2023) (Jan. 2019) (cit. on pp. 11, 16).
6. Korotkevich, G. *et al.* *Fast gene set enrichment analysis* en. Pages: 060012 Section: New Results. Feb. 2021. <https://www.biorxiv.org/content/10.1101/060012v3> (2023) (cit. on p. 16).
7. Porcu, E. *et al.* Differentially expressed genes reflect disease-induced rather than disease-causing changes in the transcriptome. en. *Nature Communications* **12**. Number: 1 Publisher: Nature Publishing Group, 5647. ISSN: 2041-1723. <https://www.nature.com/articles/s41467-021-25805-y> (2023) (Sept. 2021) (cit. on p. 19).
8. Çalışkan, M. *et al.* Exome sequencing reveals a novel mutation for autosomal recessive non-syndromic mental retardation in the *TECR* gene on chromosome 19p13. eng. *Human Molecular Genetics* **20**, 1285–1289. ISSN: 1460-2083 (Apr. 2011) (cit. on p. 19).
9. Lonsdale, J. *et al.* The Genotype-Tissue Expression (GTEx) project. en. *Nature Genetics* **45**. Number: 6 Publisher: Nature Publishing Group, 580–585. ISSN: 1546-1718. <https://www.nature.com/articles/ng.2653> (2022) (June 2013) (cit. on pp. 20, 46).

0. Oliva, M. *et al.* The impact of sex on gene expression across human tissues. *Science (New York, N.Y.)* **369**, eaba3066. ISSN: 0036-8075. <https://www.ncbi.nlm.nih.gov/pmc/articles/PMC8136152/> (2022) (Sept. 2020) (cit. on pp. 20, 46).
1. Sone, K. *et al.* Changes of estrous cycles with aging in female F344/n rats. eng. *Experimental Animals* **56**, 139–148. ISSN: 1341-1357 (Apr. 2007) (cit. on pp. 20, 46).
2. Gelman, A. *et al.* Bayesian Workflow. *arXiv:2011.01808 [stat]*. arXiv: 2011.01808. <http://arxiv.org/abs/2011.01808> (2022) (Nov. 2020) (cit. on p. 22).
3. Kruschke, J. K. Bayesian Analysis Reporting Guidelines. en. *Nature Human Behaviour* **5**. Number: 10 Publisher: Nature Publishing Group, 1282–1291. ISSN: 2397-3374. <https://www.nature.com/articles/s41562-021-01177-7> (2023) (Oct. 2021) (cit. on p. 22).
4. Kubowicz, P. *pkubowicz/opendetex* original-date: 2016-02-19T21:26:23Z. Aug. 2023. <https://github.com/pkubowicz/opendetex> (2023) (cit. on p. 29).
5. Benoit, K. *et al.* quanteda: An R package for the quantitative analysis of textual data. en. *Journal of Open Source Software* **3**, 774. ISSN: 2475-9066. <https://joss.theoj.org/papers/10.21105/joss.00774> (2023) (Oct. 2018) (cit. on p. 29).
6. Higham, N. J. Computing the nearest correlation matrix—a problem from finance. *IMA Journal of Numerical Analysis* **22**, 329–343. ISSN: 0272-4979. <https://doi.org/10.1093/imanum/22.3.329> (2022) (July 2002) (cit. on p. 34).
7. Bates, D. *et al.* *Matrix: Sparse and Dense Matrix Classes and Methods* Mar. 2022. <https://CRAN.R-project.org/package=Matrix> (2022) (cit. on p. 34).
8. Gay, N. R. *et al.* Impact of admixture and ancestry on eQTL analysis and GWAS colocalization in GTEx. *Genome Biology* **21**, 233. ISSN: 1474-760X. <https://doi.org/10.1186/s13059-020-02113-0> (2023) (Sept. 2020) (cit. on p. 41).
9. Wheeler, H. E. *et al.* *Survey of the Heritability and Sparsity of Gene Expression Traits Across Human Tissues* en. Tech. rep. Section: New Results Type: article (bioRxiv, Mar. 2016), 043653. <https://www.biorxiv.org/content/10.1101/043653v2> (2022) (cit. on p. 42).
10. Frankish, A. *et al.* GENCODE 2021. eng. *Nucleic Acids Research* **49**, D916–D923. ISSN: 13624962 (Jan. 2021) (cit. on p. 45).
11. Smith, J. R. *et al.* The Year of the Rat: The Rat Genome Database at 20: a multi-species knowledgebase and analysis platform. eng. *Nucleic Acids Research* **48**, D731–D742. ISSN: 1362-4962 (Jan. 2020) (cit. on p. 45).

0. Landen, S. *et al.* Genetic and epigenetic sex-specific adaptations to endurance exercise. eng. *Epigenetics* 14, 523–535. ISSN: 1559-2308 (June 2019) (cit. on p. 46).
1. Landen, S. *et al.* Physiological and molecular sex differences in human skeletal muscle in response to exercise training. eng. *The Journal of Physiology* 601, 419–434. ISSN: 1469-7793 (Feb. 2023) (cit. on p. 46).
2. Many, G. M. *et al.* Sexual dimorphism and the multi-omic response to exercise training in rat subcutaneous white adipose tissue. eng. *bioRxiv: The Preprint Server for Biology*. <https://www.biorxiv.org/content/10.1101/2023.02.03.527012v1> (Feb. 2023) (cit. on p. 46).
3. McConell, G. K., Wadley, G. D., Le plastrier, K. & Linden, K. C. Skeletal muscle AMPK is not activated during 2 h of moderate intensity exercise at ~65% V'O₂peak in endurance trained men. *The Journal of Physiology* 598, 3859–3870. ISSN: 0022-3751. <https://www.ncbi.nlm.nih.gov/pmc/articles/PMC7540472/> (2023) (Sept. 2020) (cit. on p. 47).
4. Neuwirth, E. *RColorBrewer: ColorBrewer Palettes* Apr. 2022. <https://cran.r-project.org/web/packages/RColorBrewer/index.html> (2023) (cit. on p. 48).
5. International Multiple Sclerosis Genetics Consortium *et al.* Genetic risk and a primary role for cell-mediated immune mechanisms in multiple sclerosis. eng. *Nature* 476, 214– 219. ISSN: 1476-4687 (Aug. 2011) (cit. on p. 51).

REVIEWER COMMENTS

Reviewer #1 (Remarks to the Author):

The authors have suitably addressed my previous concerns. I have nothing additional to add to this interesting work.

Reviewer #2 (Remarks to the Author):

Overall the manuscript is much improved. In general, the manuscript performs extensive, complex analysis for what in the end seems to be a fairly weak signal. The ability to find signal in this data is a testament to the skill of the authors. Several statements in the revision are more measured, (e.g. disease-like or less disease-like is an improvement), but we believe that additional statements about the weakness of the signal should be addressed.

Major

Statements of effect sizes / enrichment.

We appreciate the comments of reviewer 1 and the response; that was an important analysis to perform. The partitioned heritability can no longer be called significant; we would suggest the corresponding language be modified as well - e.g. "we also observed large enrichments... colon featured the strongest enrichments... proportion of heritability captured is substantial", suggest removing the 'large' and 'substantial' here and just stating the amounts.

The statement, "posterior output nevertheless reflected the circumstance where most tissue trait combinations were broadly uncertain in their directionality of enrichment, with a smaller number showing stronger confidence" is an important limitation that should be added to the manuscript.

Section is titled, "Exercise effects can override regulation of gene expression" yet this section also states, "At 8W, we observed an average of 1 (range: 0-10) genes per tissue in at least one sex with effect sizes in trained rat that were $>2SD$ the genetic component". With only 1 gene per tissue, this statement to us suggests that exercise in general does not override the genetic regulation of gene expression? Could change to "Exercise effects on regulation of gene expression".

Interpretation of results and translation. In general, I caution the authors to avoid making unsubstantiated claims that relate to translation. Human disease mechanisms and clinical medicine are very complex.

Reviewer 2 comment 1, agree that the genetic regulation of gene expression goes to mechanism is correct in the way that TWAS is usually applied. But here they are relating genetic regulation of gene expression to correlate with response to exercise (a non-genetic stimulus) in rats, and then further relate to disease. This logic / pathway is still challenging to follow. The response to the reviewer is 5 paragraphs, including "Rather, exercise stresses the body in ways not unlike disease stress (taxing the lungs and the heart), and adaptation to long-term exercise follows similar transcriptional pathways to adaptation to disease, ultimately yielding a preventative effect when the adaptive response from exercise is able to curtail disease pathogenesis," and in the manuscript, "By subjecting the body to disease-like stresses, regular exercise elicits adaptation to the symptoms of those diseases, reducing the risk of their manifestation from the disease itself". While there is extensive epidemiologic evidence for the benefits to exercise, the mechanisms are not clear. The authors statements are very broad and overarching hypotheses that require extensive validation, not statements of fact. Suggest adding "we hypothesize" or "may".

Discussion still needs to avoid unnecessary speculation that may be misinterpreted. After, "particularly intense exercise may have an attenuating effect on growth" the following sentences in discussion to do with lean mass, not height (or I suppose length, for a rat). Suggest they be removed.

Clarification of equations

If we understand correctly, in the legend of Figure 2, the term to define "genetic variance in $\log_2(\text{Gene expression})$ " is used as $SD(\text{gene expression})$ or $SD(\log_2\text{expression})$. Those terms have different forms and even are misleading in that those terms do not include the concept of genetic variance. Should SD_{geno} or SD_{pheno} that are defined in the main text should be used?

Readability

The paper is still very complex, but we appreciate the efforts at updating the text to make it more readable.

Minor:

The statement "we report on "confident" effects when a prior mass is >90% to one side of 0" - is this prior mass or posterior mass?

The abbreviation DE is used at line 127 before it is defined at line 142.

Line 521-522, for "This value was then divided by the estimated exercise DE", confirm that estimated exercise DE is numerator, not the denominator.

Dimensions for tissue, trait and trait category index in Table S2 and S3 should be 15, 99 and 12, respectively.

For Table S2 and S3, all of the notations are not for i, j or k index but for across those indexes. For example, i is defined as a vector of length 15 not for the i th tissue specifically but for across all the tissues $i=1, \dots, 15$. Please clarify.

Reviewer 2 comment 2a: thank you for including these issues in the methods.

Reviewer #3 (Remarks to the Author):

Thank you for your very thoughtful replies to each of my comments, which convinced me that you had successively revised your manuscript. Good luck on your future studies.

Dear anonymous reviewers,

We again wish to thank you for your thoughts and feedback on our *Nature Communications* submission titled “**The impact of exercise on gene regulation in association with complex trait genetics**”. We are gratified that we were able to satisfy most of your concerns during the first round of review.

Below, we attempt to address those concerns that we were not able to fully satisfy, as well as new concerns that arose during the revision process. As **Reviewers 1** and **3** did not request additional revisions, we focus the majority of this letter on point-by-point responses to the insightful comments and suggestions offered by **Reviewer 2**. In almost every case, we agreed with their suggestions and have modified the text accordingly. We have also made a small number of minor stylistic edits to individual words or label placements in figures, unremarked on explicitly but emphasized in the attached differences document. As before, you may click on the above links in **red** to proceed directly to responses corresponding to each reviewer.

We hope that these modifications were able to address lingering reservations regarding our manuscript, and are grateful for the opportunity to improve our work under your guidance.

Reviewer 1

Comment: The authors have suitably addressed my previous concerns. I have nothing additional to add to this interesting work.

Response: Thank you again for your earlier feedback on our work. Your suggestions were instrumental in improving our manuscript, and we are pleased that our revisions have met your approval.

Reviewer 2

Comment: Overall the manuscript is much improved. In general, the manuscript performs extensive, complex analysis for what in the end seems to be a fairly weak signal. The ability to find signal in this data is a testament to the skill of the authors. Several statements in the revision are more measured, (e.g. disease-like or less disease-like is an improvement), but we believe that additional statements about the weakness of the signal should be addressed.

Response: It is encouraging to hear that you have found our manuscript improved, and we thank you for identifying further areas for improvement. We agree that the text would benefit from further clarification on appropriate limits to interpretation, and so have followed your suggestions below. The effects explored here are indeed small in overall magnitude, and we would not wish readers to come away from the paper believing there to exist strong, definitive links between features that are only subtly associated.

Comment: (1.1) **Statements of effect sizes / enrichment.** We appreciate the comments of reviewer 1 and the response; that was an important analysis to perform. The partitioned heritability can no longer be called significant; we would suggest the corresponding language be modified as well - e.g. “we also observed large enrichments... colon featured the strongest enrichments... proportion of heritability captured is substantial”, suggest removing the ‘large’ and ‘substantial’ here and just stating the amounts.

The statement, “posterior output nevertheless reflected the circumstance where most tissue · trait combinations were broadly uncertain in their directionality of enrichment, with a smaller number showing stronger confidence” is an important limitation that should be added to the manuscript.

Response: We have amended our written descriptions of h^2_{SNP} enrichment results to be more explicit about the features we’d wished to highlight, limiting further interpretation to the Discussion. Specifically, we removed the written mention of the Colon’s enrichment:

“The colon, meanwhile, corresponded the strongest estimated enrichments of the Anthropometric category, specifically in the two *birth weight* traits.”

and changed

“We also observed large enrichments in the spleen, especially across various immune diseases, including asthma and multiple sclerosis.”

to:

“Across the 43 traits with at least one significant enrichment at Bonferroni-adjusted $\alpha = 0.05$, the largest significant enrichment factor corresponded most often to the spleen ($22 / 43 \approx 51\%$), especially in *Blood* ($9 / 14$) and *Immune* ($6 / 7$) phenotypes, with an average enrichment factor of ≈ 2.85 across significant enrichments.”

and

“The proportion of heritability captured by these gene sets is substantial, on the order of $\approx 10\%$ (Fig. S2a) and...”

to

“The proportion of heritability captured by these gene sets is on the order of $\approx 10\%$ (Fig. S2a), and...”

Regarding our description of trait \times tissue posterior directionality, we have modified Figure 4 to include a version of the histogram (sub-panel b) presented in our response:

modifying its caption to include:

“In b), we plot the histogram of posterior masses > 0 for trait \times tissue difference effects, with colors drawn from cell labels in a).”

and changing both caption and in-text sub-panel mentions (eg Fig. 4b, Fig. 4d, etc.) accordingly (see differences document). Additionally, we added the following sentence to our Results:

“At the trait \times tissue level, posterior output were broadly uncertain in most pairs’ directionality of enrichment, with a smaller number showing stronger confidence in positive enrichment (Fig. 4b).”

and the following passage into our discussion:

“Overall, estimates for enrichments and depletions in both intersect and directionality effects were small, even for confidently non-zero effects, predominately varying within 0 ± 0.3 on the log-odds scale (Fig. 4e-g, Fig. 5). This corresponds to a maximum difference of $\approx 7.5\%$ on the probability scale ($\text{inv-logit}(0.15) - \text{inv-logit}(-0.15)$), and is consistent with the relatively small deviations observed from the 1-to-1 lines in Fig. 4c-d. Interpretation of these exercise biological findings should not lose sight of this context: small, subtle, but nevertheless discernible association.”

Comment: (1.2) Section is titled, “Exercise effects can override regulation of gene expression” yet this section also states, “At 8W, we observed an average of 1 (range: 0-10) genes per tissue in at least one sex with effect sizes in trained rat that were $>2SD$ the genetic component”. With only 1 gene per tissue, this statement to us suggests that exercise in general does not override the genetic regulation of gene expression? Could change to “Exercise effects on regulation of gene expression”.

Response: We have made the requested change. A limitation there was our disregard of genes whose SNP-heritability could not be estimated to be above 0 at IHW- $\alpha = 0.1$. Including these, we had the opposite problem, where small heritability implied small genetic variance implied large relative effect, which was why we prioritized discussing phenotypic variance in subsequent figures and text.

Comment: (1.3) **Interpretation of results and translation.** In general, I caution the authors to avoid making unsubstantiated claims that relate to translation. Human disease mechanisms and clinical medicine are very complex. Reviewer 2 comment 1, agree that the genetic regulation of gene expression goes to mechanism is correct in the way that TWAS is usually applied. But here they are relating genetic regulation of gene expression to correlate with response to exercise (a non-genetic stimulus) in rats, and then further relate to disease. This logic / pathway is still challenging to follow. The response to the reviewer is 5 paragraphs, including “Rather, exercise stresses the body in ways not unlike disease stress (taxing the lungs and the heart), and adaptation to long-term exercise follows similar transcriptional pathways to adaptation to disease, ultimately yielding a preventative effect when the adaptive response from exercise is able to curtail disease pathogenesis,” and in the manuscript, “By subjecting the body to disease-like stresses, regular exercise elicits adaptation to the symptoms of those diseases, reducing the risk of their manifestation from the disease itself”. While there is extensive epidemiologic evidence for the benefits to exercise, the mechanisms are not clear. The authors statements are very broad and overarching hypotheses that require extensive validation, not statements of fact. Suggest adding “we hypothesize” or “may”.

Response: Thank you for the further clarification and thoughts. We agree that language around our interpretation of these results would benefit from being more explicit on the speculative nature of that interpretation. We have modified the quoted sentence from the manuscript to read:

“Finally, even where exercise regulates gene expression in ostensibly ‘disease-like’ directions, it may be that many phenotypes as those above manifest when inflammatory, hunger-regulating, or other effects of exercise occur without having first been induced by exercise. We hypothesize that by subjecting the body to disease-like stresses, regular exercise elicits adaptation to the symptoms of those diseases, reducing the risk of their manifestation from the disease itself. In this light, the presence of their signal here may also be expected.”

Otherwise, our understanding is that the use of instrumental variables such as genetic variants for causal inference does not inherently requires that later interventions on the exposure / endogenous variable (gene expression) involve mechanisms that flow through those instrumental variables. Analogously, Wright’s original use of a rainfall instrument to estimate the effects of edible oil price on demand did not require that later interventions on price (eg tariffs) involve a rainfall intermediary.

We also apologize for the length of our earlier Response.

Comment: (1.4) Discussion still needs to avoid unnecessary speculation that may be misinterpreted. After, “particularly intense exercise may have an attenuating effect on growth” the

“Evidence for exercise effects on height is weak and ambiguous. However, particularly intense exercise may have an attenuating effect on growth⁴⁶⁻⁴⁸, especially under nutritional stress. In the context of exercise training in MoTrPAC rats, males lost fat mass but not lean mass, and females gained lean mass but not fat mass. However, unexercised control female rats gained comparable lean mass, as well as fat mass, so exercise appeared to have an overall suppressive effect on growth in both sexes. Coupled with hypothetical nutritional stress during development, an overall inhibitory effect on gene-regulatory drivers of stature from exercise-like stress could be unsurprising.”

following sentences in discussion to do with lean mass, not height (or I suppose length, for a rat). Suggest they be removed.

Response: Statural variation primarily results from differences in skeletal morphology, and we agree that the shape, size, and form of bones is different from the gross body composition measurements described in text, but we do not think that their partial comparison in this context is too speculative. However, the matter is further complicated by the contradicting associations with BF% (Fig. 5), and we agree that mismatch in relevant developmental period, species, and nutritional circumstance render appropriately nuanced interpretation of limited value for this paper. We have modified the text according to your suggestion, changing:

“Evidence for exercise effects on height is weak and ambiguous. However, particularly intense exercise may have an attenuating effect on growth⁴⁶⁻⁴⁸, especially under nutritional stress, which may partially underlie the associations observed here.”

Comment: (1.5) **Clarification of equations.** If we understand correctly, in the legend of Figure 2, the term to define “genetic variance in $\log_2(\text{Gene expression})$ ” is used as $\text{SD}(\text{gene expression})$ or $\text{SD}(\log_2\text{expression})$. Those terms have different forms and even are misleading in that those terms do not include the concept of genetic variance. Should SD_{geno} or SD_{pheno} that are defined in the main text should be used?

Response: We apologize for the inconsistency. We have modified the written Methods to clarify our use of these terms, changing:

“For each gene, we then took the square root of the posterior mean of inferred expression variance to estimate within-population standard deviation of gene expression. This value was then divided by the estimated exercise DE to produce standardized estimates in units of within-tissue phenotypic standard deviation (SD_{pheno}).”

to:

“For each gene, we then took the square root of the posterior mean of inferred $\sqrt{\text{log}_2\text{expression variance (} \text{Var}(\log_2(\text{gene expression}))\text{)}$ to estimate within-population standard deviation of the magnitude of gene expression. We then divided estimated exercise DE by these values to produce standardized estimates in units of within-tissue phenotypic standard deviation (SD_{pheno}).”

also correcting the error identified in a later **Comment** (2.3).

We have additionally corrected the axis labels in Fig. 2b:

and the corresponding figure caption:

“In b), we plot the empirical quantile function for distributions of ratios of each tissue’s exercise-induced log₂(gene expression) / SD(log₂(gene expression)).”

to:

Comment: (1.6) **Readability.** The paper is still very complex, but we appreciate the efforts at updating the text to make it more readable.

Response: Thank you for the affirmation. We are happy to have improved the text’s readability following your suggestions.

Comment: (2.1) The statement “we report on ”confident” effects when a prior mass is >90% to

Response: Thank you for catching that mistake. We did indeed mean to write *posterior* mass one side of 0” - is this prior mass or posterior mass? and have changed:

“...we report on ”confident“ effects when a prior mass is >90% to one side of 0.”

“...we report on ”confident“ effects when a posterior mass is >90% to one side of 0.”

Comment: (2.2) The abbreviation DE is used at line 127 before it is defined at line 142.

Response: We have corrected this error to define the DE abbreviation the first time it is used, amending:

“...many of these genes can be mapped to human orthologs: 94.5% of all unique, (DE) genes (87-98% across tissues)...”

to:

“...many of these genes can be mapped to human orthologs: 94.5% of all unique, differentially expressed (DE) genes (87-98% across tissues)...”

and removing the now superfluous later definition.

Comment: (2.3) Line 521-522, for “This value was then divided by the estimated exercise DE”, confirm that estimated exercise DE is numerator, not the denominator.

Response: Thank you for catching this error. We have changed the corresponding text to correctly locate exercise DE in the numerator (see **Response** to **Comment** 1.5).

Comment: (2.4) Dimensions for tissue, trait and trait category index in Table S2 and S3 should be 15, 99 and 12, respectively.

Response: Upon careful consideration, our understanding is that the text is correct as written. Specifically, i , j , and k are index variables stored in one-dimensional integer arrays of length 1485. For example, the variable i may look like [1, 1, 1, 1, ..., 2, 2, 2, ...1, or [7, 15, 2, 1, 1, 13, ...1, depending on how these data were entered. It has 1485 positions that may take on positive integer values between 1 and 15 inclusive. Technically, these are not all independently free to vary: depending on the order in which the array is filled, i may have an effective “dimensionality” between 1386 and 1484 (if you know the first 1484 values, you will always know the 1485th, as there can only be one possible value remaining to fill that position given the constraints of the index. If you fill the array in order, then the final 99 values must all be 15).

As such, i can be thought of as a discrete random variable generated by sampling without replacement from a pool of 99 ‘1’s, 99 ‘2’s, 99 ‘3’s, etc. Alternatively, it can be considered as a single draw from the set of all permutations of these values. If it were unobserved (eg, the tissue labels were accidentally shuffled or lost during earlier processing), we might attempt to leverage our knowledge of their structure to infer these 1485 states by eg specifying a discrete uniform prior over all permutations. It is “observed”, however, and conditioned on – we assume the indices to be known without error, just as we assume the discrete cell counts are known without error.

We do apologize for this abuse of notation, though. In part, our perspective may also stem from our implementation of the model. The data block in the corresponding *.Stan file for the first model starts:

```
data \{
  int<lower=1> row_n;
  int<lower=1> col_n;
  int<lower=1> colcat_n;
  int<lower=0> total [row_n * col_n];
  int<lower=1,upper=row_n > row_index [row_n * col_n];
  int<lower=1,upper=col_n > col_index [row_n * col_n];
  ...
}
```

Which may have influenced how we view such variables as written in the main-text’s style of notation.

Comment: (2.5) For Table S2 and S3, all of the notations are not for i, j or k index but for across those indexes. For example, i is defined as a vector of length 15 not for the i th tissue specifically but for across all the tissues $i=1, \dots, 15$. Please clarify.

Response: Please see above. We are open to changing the tables if our reasoning is in error, or perhaps omitting “observed” variables from it altogether.

Comment: (2.6) Reviewer 2 comment 2a: thank you for including these issues in the methods.

Response: Thank you for the comment and acknowledgement of our improvement! We strongly appreciate the extensive thought and expertise you have brought to bear in reviewing our manuscript.

Reviewer 3

Comment: Thank you for your very thoughtful replies to each of my comments, which convinced me that you had successively revised your manuscript. Good luck on your future studies.

Response: Thank you for the kind words and wishes! We are grateful for your assistance and feedback, and are happy to hear that you are satisfied by our revisions.

REVIEWERS' COMMENTS

Reviewer #2 (Remarks to the Author):

We appreciate the authors' efforts and the thoroughness of their responses. We have no further comments or questions.

We deeply appreciate the time and effort all reviewers dedicated to reviewing our manuscript titled “**The impact of exercise on gene regulation in association with complex trait genetics**” and submitted to *Nature Communications*. Your feedback was invaluable in enhancing the quality and clarity of our work.

Below, we reproduce verbatim all comments provided during this round of review, followed by our response:

Reviewer 2

Comment: We appreciate the authors’ efforts and the thoroughness of their responses. We have no further comments or questions.

Response: Thank you for the acknowledgement, and for your own diligence as a reviewer. We are happy to have satisfied your concerns.